# Widespread transcript shortening through alternative polyadenylation in secretory cell differentiation

Larry C. Cheng [1,2,3], Dinghai Zheng [2], Erdene Baljinnyam[2], Fangzheng Sun[2], Koichi Ogami[4,6], Percy Luk Yeung[5], Mainul Hoque[2], Chi-Wei Lu[5], James L. Manley[4] & Bin Tian [1,2,3✉]

Most eukaryotic genes produce alternative polyadenylation (APA) isoforms. Here we report that, unlike previously characterized cell lineages, differentiation of syncytiotrophoblast (SCT), a cell type critical for hormone production and secretion during pregnancy, elicits widespread transcript shortening through APA in 3'UTRs and in introns. This global APA change is observed in multiple in vitro trophoblast differentiation models, and in single cells from placentas at different stages of pregnancy. Strikingly, the transcript shortening is unrelated to cell proliferation, a feature previously associated with APA control, but instead accompanies increased secretory functions. We show that 3'UTR shortening leads to transcripts with higher mRNA stability, which augments transcriptional activation, especially for genes involved in secretion. Moreover, this mechanism, named secretion-coupled APA (SCAP), is also executed in B cell differentiation to plasma cells. Together, our data indicate that SCAP tailors the transcriptome during formation of secretory cells, boosting their protein production and secretion capacity.

[1] Graduate Program in Quantitative Biomedicine, School of Graduate Studies, Rutgers University, New Brunswick, NJ 08901, USA. [2] Department of Microbiology, Biochemistry and Molecular Genetics, Rutgers New Jersey Medical School, Newark, NJ 07103, USA. [3] Program in Gene Expression and Regulation, and Center for Systems and Computational Biology, Wistar Institute, Philadelphia, PA 19104, USA. [4] Department of Biological Sciences, Columbia University, New York, NY 10027, USA. [5] Robert Wood Johnson Medical School and Child Health Institute of New Jersey, New Brunswick, NJ 08901, USA. [6]Present address: Department of Biological Chemistry, Graduate School of Pharmaceutical Sciences, Nagoya City University, Nagoya 467-8603, Japan. ✉email: btian@wistar.org

The placenta functions as the lung and kidney of the fetus and produces hormones to maintain the pregnancy[1,2]. Derived from the trophectoderm layer of blastocyst, trophoblasts (TBs) are the main cell types that form the outer layer of placenta[3]. Among the three TB subtypes, proliferating villous cytotrophoblasts (VCTs) form the germative layer that gives rise to the other two types, extravillous trophoblasts (EVTs) and syncytiotrophoblasts (SCTs)[3]. EVTs invade the decidua to anchor the placenta to uterus and modify maternal vasculature[1], whereas SCTs form the interface between the mother and fetus, where nutrient/waste exchange takes place[1], and secrete a variety of hormones necessary for pregnancy, such as human chorionic gonadotropin (hCG), human placental lactogen, human placental growth hormone, progesterone, and estradiol[4]. While transcriptional regulation in TB differentiation has been extensively studied[5–7], little is known about post-transcriptional control of gene expression in the process.

Almost all eukaryotic messenger RNAs (mRNAs) require cleavage and polyadenylation for 3′ end maturation[8,9]. Over 70% of mammalian genes harbor multiple polyadenylation sites (PASs), leading to expression of alternative polyadenylation (APA) isoforms[10,11]. Most APA sites are located in the 3′ most exon, resulting in isoforms with different 3′UTR sizes[11]. 3′UTR APA can play a substantial role in gene expression through regulation of sequence or structure motifs that impact aspects of mRNA metabolism, such as translation, stability and localization[12–14]. In addition, about 20% of human genes have APA sites located in introns, which additionally change coding sequences when used[11,15,16].

The relative abundance of an APA isoform can vary widely across tissue types[17]. For example, brain tissues tend to express long 3′UTR isoforms, whereas blood and testis show the opposite trend[16–19]. In addition, increased cell proliferation rate has been associated with shortening of 3′UTRs through APA[20], while 3′UTRs lengthen during cell differentiation and development[21,22]. Here, we report the unexpected finding of widespread transcript shortening through APA, including 3′UTR shortening and activation of intronic polyadenylation (IPA), during differentiation of SCTs. We examine several in vitro models as well as single placental cells in vivo. We further generalize this finding to B-cell differentiation.

## Results

**Trophoblast differentiation displays a distinct APA profile.** To study APA regulation in different cell differentiation lineages, we analyzed RNA-seq data previously generated by Xie et al.[23], involving differentiation of human embryonic stem cells (hESCs, H1 line) into neuronal projector cells, mesenchymal stem cells, mesendoderm, and TBs (illustrated in Fig. 1a)[23]. By comparing RNA-seq reads covering different portions of 3′UTR that were subject to APA regulation (illustrated in Fig. 1b and see "Methods" for details), we found that, consistent with previous reports[24,25], differentiation of hESCs into neuronal projector cells, mesenchymal stem cells, or mesendoderm led to 3′UTR lengthening to variable degrees (Fig. 1c). By contrast, surprisingly, differentiation of hESCs into TBs showed 3′UTR shortening (Fig. 1c).

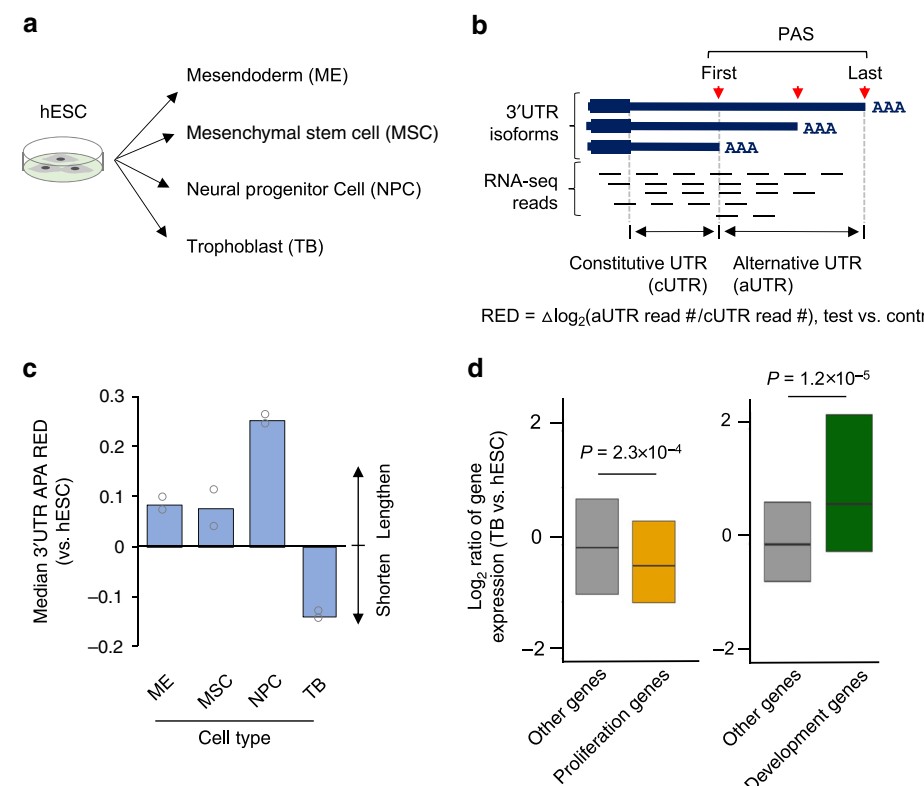

**Fig. 1 Trophoblast differentiation displays a unique APA profile. a** Schematic showing differentiation of human embryonic stem cells (hESCs, H1 line) into four lineages (a study by Xie et al.[23]). **b** Schematic of 3′UTR APA analysis using RNA-seq data. RNA-seq reads in 3′UTRs are divided into constitutive UTR (cUTR) and alternative UTR (aUTR) groups based on the first and last PASs, as indicated. 3′UTR length change for a gene is indicated by relative expression difference (RED) value between two samples or sample groups, as shown in the graph. RED is calculated as indicated. **c** Median REDs of four differentiation lineages as compared to hESC. A total of 8,712 genes were used for analysis. **d** Boxplots showing expression changes of proliferation genes (left; 350 genes; defined by Sandberg et al.[20]) and development genes (right; 74 genes; defined by Ji et al.[21]) in TB vs. hESC. P-values (Wilcoxon test) based on comparisons with other genes are indicated.

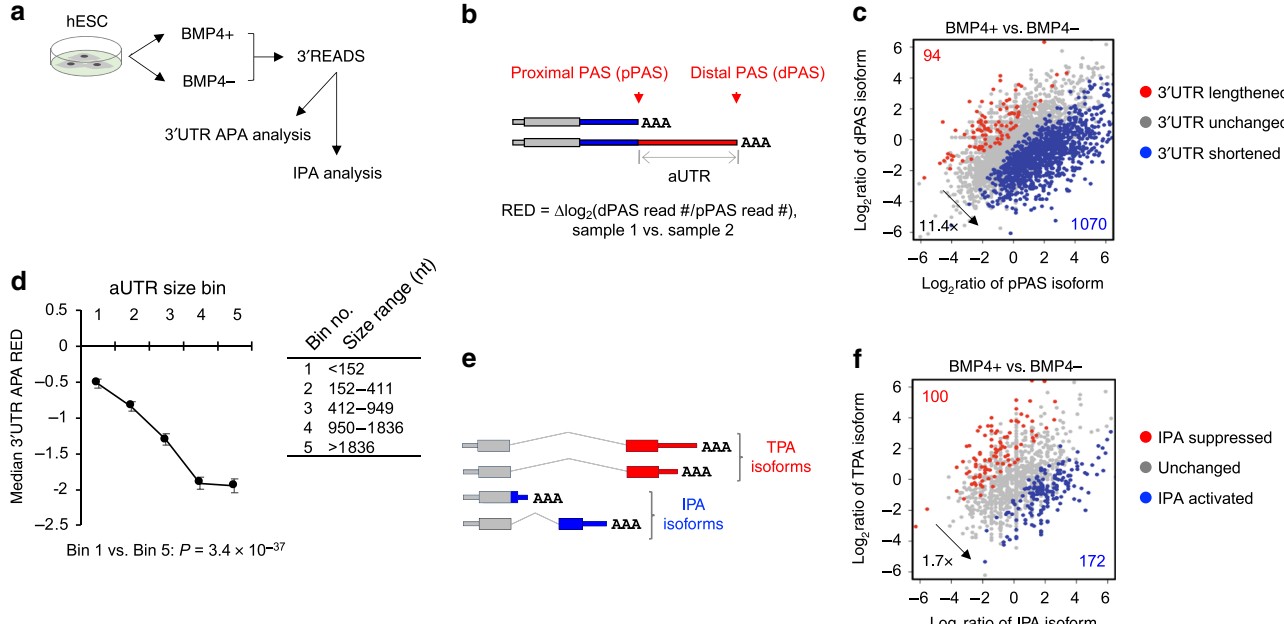

**Fig. 2 Widespread 3′UTR shortening and IPA activation in trophoblast differentiation. a** Schematic of 3′READS experiment using hESCs (H9 line) treated with BMP4 for TB differentiation. **b** Schematic of 3′UTR APA analysis using 3′READS data. Two representative 3′UTR isoforms are selected, named proximal and distal PAS (pPAS and dPAS) isoforms, respectively. RED is calculated as indicated. **c** Scatter plot showing 3′UTR APA changes. Each dot is a gene with two selected 3′UTR isoforms. The numbers of genes showing significantly lengthened 3′UTRs (red) or shortened 3′UTRs (blue) are indicated, and so is the ratio of these two numbers. Significance here and other panels of this figure is based on $P < 0.05$ (Fisher's exact test) and > 20% change of expression ratio of the two isoforms. **d** Relationship between aUTR size and degree of 3′UTR size regulation (represented by RED). Genes are grouped into five similarly sized bins (about 500 genes each) based on their aUTR length. aUTR size range for each bin is shown in a table. Data are presented as median value ± SEM. *P*-value (Wilcoxon test) comparing Bins 1 and 5 is indicated. **e** Schematic of intronic polyadenylation (IPA) isoforms and 3′ terminal exon polyadenylation (TPA) isoforms. **f** Scatter plot showing IPA changes. Each dot is a gene with expression of both IPA and TPA isoforms. The numbers of genes with significant IPA suppression (red) or IPA activation (blue) are indicated, and so is their ratio.

Increased cell proliferation could trigger 3′UTR shortening[20]. Interestingly, the proliferation genes that were previously found to be negatively correlated with 3′UTR size[20] were in fact globally downregulated during TB differentiation (Fig. 1d, left). Note that the same genes did indicate decreased cell proliferation in differentiation of mouse myoblast cell line C2C12 (Supplementary Fig. 1a), a model we have used to study 3′UTR lengthening in cell differentiation (Supplementary Fig. 1b)[21,26]. We previously identified a gene set whose expression levels positively correlated with 3′UTR size in mouse embryonic development[21]. While expression of this development gene set significantly increased in C2C12 differentiation (Supplementary Fig. 1a), they were also upregulated in hESC to TB differentiation (Fig. 1d, right), again indicating that 3′UTR size regulation in TB differentiation is different than previously studied differentiation and development systems.

**Widespread transcript shortening in TB differentiation.** To examine APA in TB differentiation in greater detail, we differentiated hESCs (H9 line) to TBs through the BMP4 induction protocol[27], and subjected cellular RNA to 3′ Region Extraction and Deep Sequencing (3′READS), a deep-sequencing method we previously developed to specifically interrogate 3′ ends of poly(A) + RNAs[11] (Fig. 2a). Based on the top two most abundant 3′UTR APA isoforms of each gene (Fig. 2b), we found that genes showing increased relative expression of short 3′UTR isoform greatly outnumbered those showing increased relative expression of long 3′UTR isoform by 11.4-fold in TBs vs. hESCs (1070 vs. 94, Fig. 2c), indicating a substantial and global trend of 3′UTR shortening. Based on the size of the 3′UTR sequence that is different between the two isoforms, named alternative UTR or

aUTR (illustrated in Fig. 2b), we divided genes into five bins (Fig. 2d). Using relative expression difference (RED) to represent the degree of 3′UTR APA regulation (described in Fig. 2b), we found that genes tended to display a greater extent of 3′UTR shortening as aUTR size increased (Fig. 2d). For example, genes with an aUTR size >1836 nucleotides (nt, bin 5) showed much more conspicuous 3′UTR shortening than genes with an aUTR size <152 nt (bin 1, $P = 3.4 \times 10^{-37}$, Wilcoxon test, Fig. 2d).

A sizable fraction of APA sites are located in introns[11], leading to IPA isoforms that have coding sequence changes in addition to 3′UTR alternations (illustrated in Fig. 2e). Using isoforms containing PASs in the 3′ terminal exons as reference, we found that IPA isoforms were generally upregulated in TBs vs. hESCs, with 1.7-fold more genes showing IPA activation than those showing suppression (172 vs. 100, Fig. 2f). Gene Ontology (GO) analysis indicated that the most significant biological processes associated with genes showing 3′UTR shortening appeared to be related to protein and RNA metabolism (Table 1), such as "regulation of catabolic process", "post-transcriptional gene regulation", and "protein localization to organelle". For genes showing IPA activation (Table 2), "regulation of mRNA catabolic process," "cellular response to acid chemical", and "cytoplasmic translational initiation" were the top enriched biological processes. Altogether, the 3′READS result corroborated our initial finding of global 3′UTR shortening in TB differentiation, and additionally revealed IPA activation. These APA events could potentially impact a diverse array of biological processes.

**APA regulation correlates with syncytiotrophoblast formation.** As there are three TB subtypes, namely, VCT, EVT, and SCT, we next asked which TB subtype was responsible for APA regulation

**Table 1 Top biological processes enriched for genes that display 3′UTR shortening in TB differentiation from hESCs.**

| P-value | Biological process |
|---|---|
| 1.5E-06 | Regulation of catabolic process. |
| 2.7E-06 | Post-transcriptional regulation of gene expression. |
| 3.5E-05 | Protein localization to organelle. |
| 6.1E-05 | Membrane organization. |
| 1.9E-04 | Regulation of mRNA metabolic process. |
| 1.9E-04 | Nucleus organization. |
| 2.4E-04 | Positive regulation of transport. |
| 2.5E-04 | Nuclear pore organization. |
| 2.6E-04 | Endomembrane system organization. |
| 3.0E-04 | Symbiosis, encompassing mutualism through parasitism. |

**Table 2 Top biological processes enriched for genes that display upregulation of IPA isoforms in TB differentiation from hESCs.**

| P-value | Biological process |
|---|---|
| 6.7E-05 | Regulation of mRNA catabolic process. |
| 6.1E-04 | Cellular response to acid chemical. |
| 2.4E-03 | Cytoplasmic translational initiation. |
| 2.4E-03 | Oligosaccharide-lipid intermediate biosynthetic process. |
| 2.6E-03 | Gliogenesis. |
| 4.0E-03 | Protein–DNA complex subunit organization. |
| 4.2E-03 | Myelination. |
| 5.1E-03 | Cell morphogenesis involved in differentiation. |
| 8.1E-03 | Regulation of translational initiation. |
| 8.1E-03 | Response to monosaccharide. |

when differentiated from hESCs. Using gene expression datasets from four TB subtype studies[28–31], we constructed a panel of 69 genes as biomarkers for three TB subtypes (Fig. 3a, and "Methods" for detail), with each subtype having one third of the genes (see Supplementary Table 1 for the full list). Using these marker genes, we found that differentiation of hESCs to TBs by BMP4 treatment was biased toward the SCT lineage (Fig. 3b), suggesting that SCT differentiation might be chiefly responsible for the observed APA regulation.

A previous study by Yabe et al.[32] separated hESC-derived TBs (induced by BMP4/A83-01/PD173073 or BAP) into three populations based on cell size, namely, < 40 μm, between 40 μm and 70 μm, and >70 μm (illustrated in Fig. 3c). As such, larger cells were more likely to be syncytialized SCTs than smaller cells. Indeed, using the RNA-seq data from the study and our TB subtype marker genes, we confirmed this notion (Supplementary Fig. 2a) and, importantly, found that 3′UTR sizes decreased as cell size increased (Fig. 3d). The largest cells (>70 μm) displayed significantly shorter 3′UTRs than did the smallest cells (< 40 μm) ($P < 0.05$, t-test, Fig. 3d), confirming that differentiation of SCTs elicits 3′UTR shortening. Note that while the three hESC to TB differentiation datasets used here involved different hESC lines and induction protocols, leading to differences in gene expression changes (Supplementary Fig. 2b), all three converged on global 3′UTR shortening, highlighting the robustness of our finding.

In addition to the hESC differentiation model, human choriocarcinoma cell line BeWo is a well-established system for SCT formation[33,34]. Using RNA-seq data generated by Azar et al.[34], in which BeWo cells were induced to fuse by forskolin (Fsk)(illustrated in Fig. 3e), we found that, as expected, SCT marker genes were significantly upregulated after two days of Fsk treatment (Supplementary Fig. 3a). Importantly, 3′UTRs

significantly shortened at the same time point (Fig. 3f). The extent of 3′UTR shortening in the BeWo model appeared similar to that in the hESC model, as indicated by their median 3′UTR APA REDs (−0.15 vs. −0.17, Fig. 3d, f) and RED distributions (Supplementary Fig. 3b). In addition, the RNA-seq data-derived 3′UTR APA REDs from the BeWo model and the hESC model were much better correlated for the genes that showed 3′UTR shortening in the 3′READS data ($r = 0.40$, blue dots in Fig. 3g) than for other genes containing 3′UTR APA sites ($r = 0.26$, gray dots in Fig. 3g), further indicating consistency between the two SCT formation models. Both 3′READS and RNA-seq data for an example gene *DNAJC3* (encoding DNAJ heat shock protein family member C3) are shown in Fig. 3h, where REDs and P-values highlight its consistent and significant 3′UTR shortening in all the models.

Using Fsk-induced BeWo cell fusion model (Supplementary Fig. 3c), we confirmed upregulation of SCT marker genes, *CGB* and *ERVFRD-1* (Supplementary Fig. 3d) by real-time quantitative PCR (RT-qPCR). In addition, using primer sets targeting different APA isoforms (illustrated in Supplementary Fig. 3e, top, and Supplementary Table 2), we confirmed 3′UTR shortening of a number of genes that displayed significant 3′UTR shortening in the RNA-seq data, such as *DNAJC3*, *PLEKHA6*, *SPCS3*, and *TIMP2* (Supplementary Fig. 3e, bottom).

We also examined a mouse model of TB differentiation, in which ectopic expression of a constitutively active *Hras* mutant $Hras^{Q61L}$ in mouse ESCs led to formation of syncytial giant cells[35]. Using 3′READS (three biological replicates, Supplementary Fig. 4a), we found that $Hras^{Q61L}$ expression in mouse ESCs elicited both global 3′UTR shortening (a 8.3-fold bias in gene number between shortened and lengthened genes, Supplementary Fig. 4b) and IPA activation (a 20.9-fold bias in gene number, Supplementary Fig. 4c). Note that while the mESC model did not involve upregulation of human SCT subtype marker genes (Supplementary Fig. 4d) or development genes (Supplementary Fig. 4e), cell proliferation genes were slightly downregulated (Supplementary Fig. 4e). These results indicate that despite many differences between human and mouse TB models, they both display global 3′UTR shortening and IPA activation.

**Single-cell analysis in vivo corroborates in vitro findings**. Several recent studies have generated single-cell RNA-seq (scRNA-seq) data from the placenta[29,30,36], creating opportunities to interrogate APA in TBs in vivo. To address read paucity in single-cell data, which could lead to high noise levels for APA analysis[25], we examined APA in different cell types using aggregated scRNA-seq data. This method, named single-cell significance analysis of APA (scSAAP, illustrated in Fig. 4a and see Methods for detail), first clustered cells based on their gene expression profiles; TB subtypes were identified using the TB subtype marker gene panel; RNA-seq reads from all cells of the same type were then combined for 3′UTR APA analysis.

Applying scSAAP to three datasets corresponding to placental cells isolated at different stages of pregnancy, including the first[30,36] and third[29] trimesters, we found that SCTs consistently displayed shorter 3′UTRs than VCTs and EVTs ($P < 0.05$, t-test, Fig. 4b). Importantly, the genes showing 3′UTR shortening in the hESC model (based on 3′READS data, Fig. 2c) displayed greater 3′UTR shortening in vivo (SCTs vs. VCTs) than other genes with 3′UTR APA sites ($P = 4.0 \times 10^{-11}$, Kolmogorov–Smirnov (K–S) test, Fig. 4c). The scRNA-seq data for *DNAJC3* is shown in Fig. 4d, which matched well with bulk RNA-seq and 3′READS data from in vitro models (Fig. 3h).

The single-cell transcriptome data could also be used to decipher relationships between cells at different differentiation

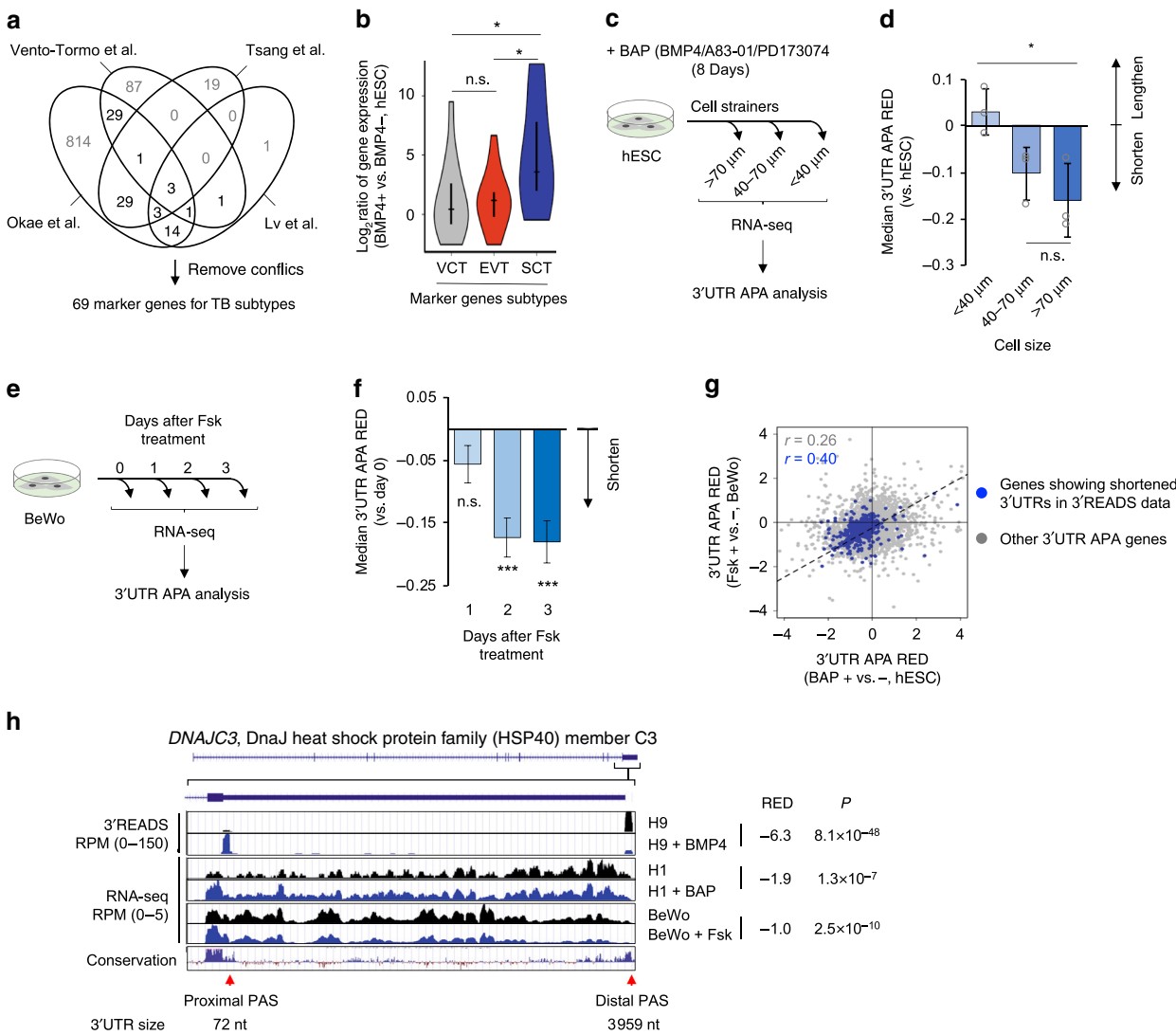

**Fig. 3 Syncytialization of trophoblasts elicits 3′UTR shortening. a** Venn diagram showing TB subtype marker genes based on four indicated publications. A panel of 69 marker genes were selected. Subtype marker genes were defined by at least two studies (gene number shown in bold) and had no conflicts across all four studies. **b** Violin plot showing expression changes of TB subtype marker genes in differentiation of hESCs into TBs by BMP4 treatment (Fig. 2a). Significance of difference between gene sets (Wilcoxon test) is indicated. **c** Schematic showing experimental design of the Yabe et al.[32] study, in which TBs differentiated from hESCs (H1 line) by BAP (BMP4, A83-01, and PD173074) were fractionated into different size groups by cell strainers. **d** Median 3′UTR APA REDs for different TB cell size groups vs. hESCs. Data are presented as mean value+/− SD of three replicates. Significance of difference (t-test) between cell groups is indicated. A total of 8216 genes were used for analysis. **e** Schematic showing experimental design of the Azar et al.[34] study, in which BeWo cells were induced to syncytialize by forskolin (Fsk). **f** Median 3′UTR APA REDs for different days of Fsk treatment vs. day 0. Data are presented as mean value+/− SD based on bootstrapped data (20 times). Significance of difference (t-test) with day 0 is indicated. A total of 7682 genes were used for analysis. **g** Scatter plot comparing 3′UTR APA REDs of hESC and BeWo RNA-seq data. Pearson correlation coefficients are indicated for genes showing 3′UTR shortening in the 3′READS data (blue dots, 1070 genes from Fig. 2c) and other genes displaying 3′UTR APA (gray dots, 7133 genes). **h** UCSC Genome Browser tracks showing data for the gene DNAJC3. 3′READS data for the hESC model (Fig. 2a), RNA-seq data for the hESC model (>70 μm TB cells vs. hESCs, Fig. 3c), and RNA-seq data for BeWo syncytialization (day 3 vs. day 0, Fig. 3e) are shown. The two APA sites and 3′UTR sizes for their corresponding isoforms are indicated. Sequence conservation based on 100 vertebrates is shown. REDs and P-values are indicated.

stages[37]. Using TB subtype marker genes, we divided cells of each TB subtype into two portions, "near" and "far", based on distance to the converged point of all subtypes (Fig. 4e). As such, cells in the far group of each subtype had higher expression levels of the corresponding marker genes, and thus could be considered more differentiated as compared to those in the near group. Interestingly, using the first trimester placental cell data from Vento-Tormo et al.[30], we found that the far group in the SCT lineage had significantly shorter 3′UTRs as compared to the near group ($P < 2.2 \times 10^{-16}$, Wilcoxon test, Fig. 4f). By contrast, the far group of VCT lineage displayed longer 3′UTRs than its near group

($P < 2.2 \times 10^{-16}$, Wilcoxon test, Fig. 4f). Since the path from VCT far cells to SCT far cells could represent the full SCT differentiation lineage in placenta, this result supports the notion that SCT differentiation in vivo, like in in vitro models, involves progressive 3′UTR shortening. This notion is also supported by the third trimester placental cell data by Tsang et al.[29] (Supplementary Fig. 5a), in which SCT far cells showed shortest 3′UTRs among all cell groups (Supplementary Fig. 5b and see Supplementary Fig. 5c for DNAJC3 example). Interestingly, the difference in 3′UTR size between VCT near and far cells was distinct from that in first trimester samples (Fig. 4f vs.

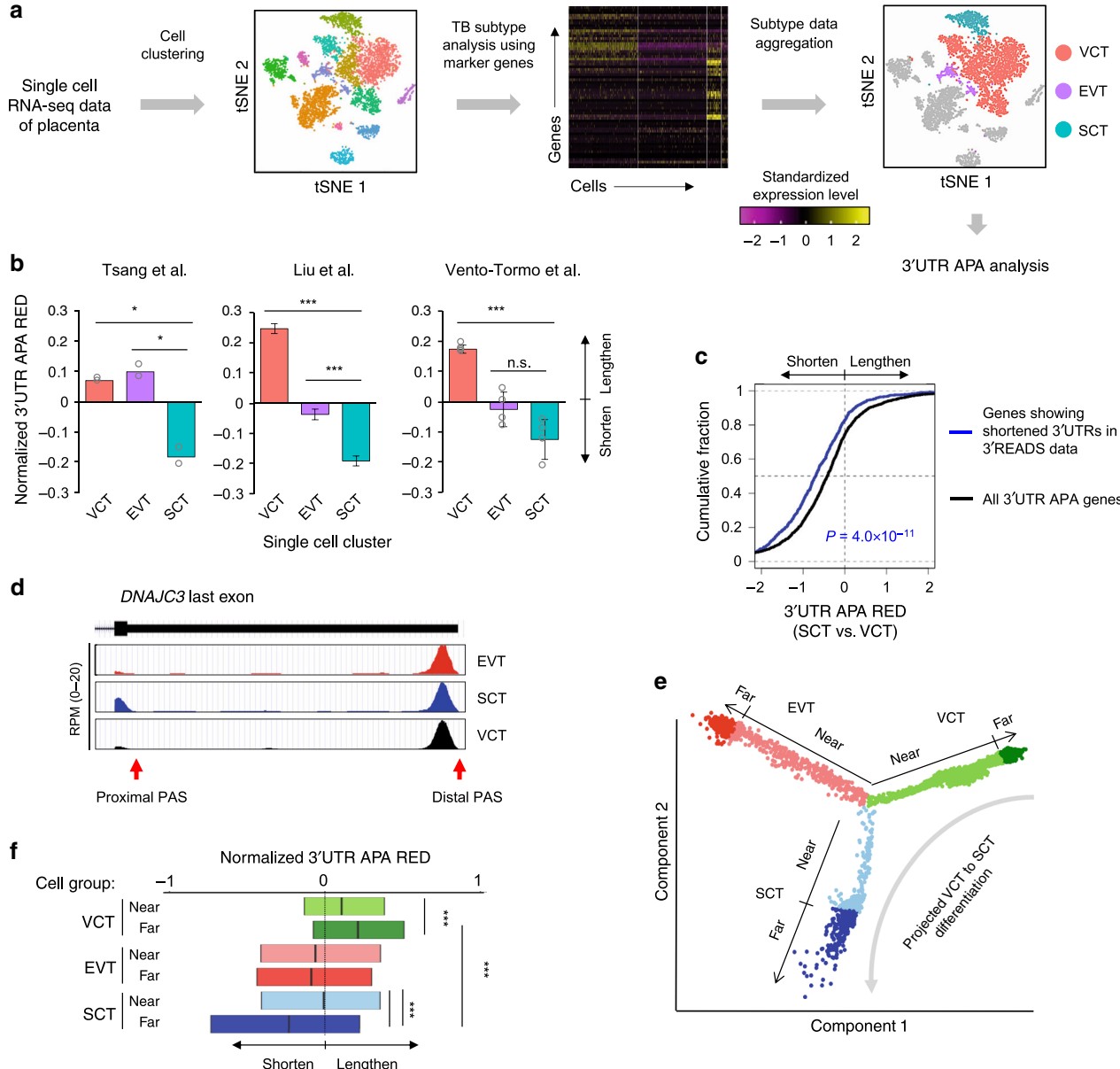

**Fig. 4 Single-cell analysis reveals short 3′UTRs in SCTs. a** Schematic of the single cell significance analysis of alternative polyadenylation (scSAAP) method. Cells are clustered by the Seurat package based on all gene expression values. The result is presented by the t-distributed stochastic neighbor embedding (tSNE) method. TB clusters are identified and grouped using the TB subtype marker gene panel. Data for each subtype are aggregated and treated as bulk RNA-seq data for 3′UTR APA analysis. **b** scSAAP analysis of placental single-cell RNA-seq datasets from three indicated studies. 3′UTR APA REDs of each dataset were normalized to the mean of all samples. Statistical significance is based on the Student's t-test. Data are presented as mean value+/− SD (two replicates in Tsang et al.[29], four replicates in Vento-Tormo et al.[30], and 20 randomly sampled data in Liu et al.[36]). **c** Cumulative distribution function (CDF) curves comparing 3′UTR APA REDs in SCTs vs. VCTs (from Vento-Tormo et al.[30] data) for genes showing 3′UTR shortening in the hESC model (blue curve, 685 genes from Fig. 2c) and all 3′UTR APA genes (black curve, 2379 genes). P-value (K–S test) for significance of difference between the two gene sets is indicated. **d** UCSC Genome Browser tracks showing single-cell data for *DNAJC3*. scRNA-seq reads are based on data from the Vento-Tormo et al.[30] study. **e** Pseudotime analysis of TB cells. The trajectories of TB cells, plotted by using DDRTree, are based on expression of TB subtype marker genes. Each lineage is divided into two halves, far and near, with similar cell numbers. Cells in each group are colored as indicated. The projected differentiation path from VCT far to SCT far is indicated. **f** 3′UTR APA REDs (4852 genes) for different TB groups, corresponding to those in **e**. 3′UTR APA REDs are normalized to the mean of all groups. Significance of RED difference (Wilcoxon test) between groups is indicated.

Supplementary Fig. 5b). Altogether, single-cell transcriptomic analysis of placental cells confirmed short 3′UTR expression in SCTs in vivo.

**APA changes are coupled to secretion gene expression.** While our analyses indicated 3′UTR shortening during SCT differentiation both in vitro and in vivo, intriguingly, we did not observe significant 3′UTR size changes during in vitro syncytialization of primary human trophoblast (PHT) cells, based on studies by Azar et al.[34] and Yabe et al.[32] (Fig. 5a). This is despite the fact that these cells underwent successful cell fusion[32,34] and upregulation of SCT marker genes (Supplementary Fig. 6), suggesting that syncytialization process per se could be uncoupled

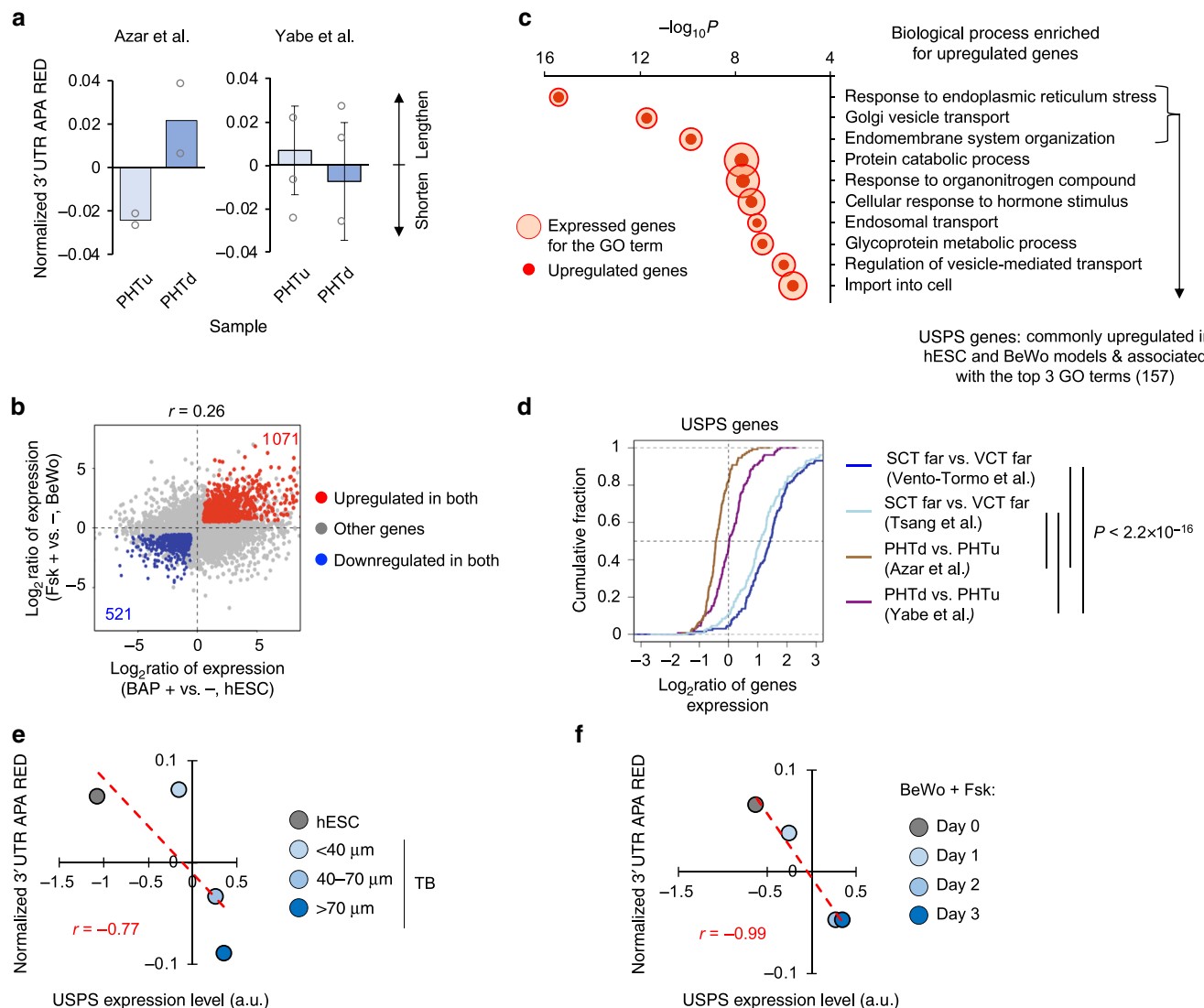

**Fig. 5 Integrative analysis connects 3′UTR size regulation to protein secretion. a** 3′UTR APA analysis of in vitro syncytialization of primary human trophoblast (PHT) cells. PHTu and PHTd are mononucleated PHT cells and syncytialized PHT cells, respectively. 3′UTR APA REDs are normalized to mean of all samples. Error bars are standard deviation based on two samples. n.s., $P \geq 0.05$ (t-test). **b** Significantly regulated genes in hESC and BeWo models. Significance of regulation is based on fold change > 1.2 and FDR = 0.05 (DESeq) in both models. The hESC model data correspond to > 70 μm TB cells vs. hESCs (Fig. 3c) and the BeWo model data correspond to Fsk + vs. Fsk- (Fig. 3e). Commonly regulated genes are highlighted. Red for upregulation and blue for downregulation. Pearson correlation coefficient (r) comparing gene regulation in the two models is indicated. **c** Bubble plot showing top GO terms (Biological Processes only) enriched for genes upregulated in both hESC and BeWo models. Each GO term is represented by a circle, whose size reflects the number of genes associated with the term, and also by a dot, whose size reflects the number of upregulated genes associated with the term. All sizes are relative to other circles in the plot. For example, the term "response to endoplasmic reticulum stress" is associated with 273 genes, of which 62 are upregulated. Definition of USPS (3′UTR size-related protein secretion) genes (157 in total) is indicated. **d** CDF curves for USPS gene expression changes in four datasets, including two in vitro syncytialization models shown in **a** and two single-cell datasets (SCT far vs. VCT far, Fig. 4). A total of 131 USPS genes with detectable expression in all datasets are used for analysis. P-values (K–S test) for significance of difference between gene sets are indicated. **e, f** Scatter plots showing correlation between 3′UTR REDs and USPS gene expression levels for the hESC model (**e**) and the BeWo model (**f**). 3′UTR APA REDs are normalized to median of all samples. a.u., arbitrary value.

from APA regulation. This prompted us to explore other biological processes that might accompany APA changes in SCT differentiation.

We found that, while similar in APA regulation, the hESC and BeWo models had substantial differences in gene expression changes, as indicated by a modest correlation coefficient ($r = 0.26$, Pearson Correlation, Fig. 5b). We thus reasoned that consistently regulated genes between these two divergent models might provide clues about some common underlying biological processes related to APA regulation. To this end, we identified

1071 upregulated genes and 521 downregulated genes common to both models (fold change > 1.2 and false discovery rate = 0.05, DESeq, Fig. 5b). GO analysis found several biological processes that were significantly enriched for upregulated genes (Fig. 5c). Notably, the top three biological processes appeared to be substantially more significant than others (all with P-value < $1.0 \times 10^{-9}$, Fisher's exact test, Fig. 5c), including "response to endoplasmic reticulum stress", "Golgi vesicle transport", and "endomembrane system organization". By contrast, Biological processes enriched for commonly downregulated genes

were much less significant based on *P*-values (Supplementary Fig. 7).

We next isolated 157 genes that were commonly upregulated in both hESC and BeWo models and were also associated with the top three GO processes. As both models involved 3′UTR shortening and the top three processes were all related to expression or transport of secreted proteins[38,39], we named these genes 3′UTR Size-related Protein Secretion (USPS) genes (listed in Supplementary Table 3). Supporting the connection between 3′UTR shortening and protein secretion, we found that USPS genes were significantly upregulated in placental SCT far cells vs. VCT far cells (both first trimester and third trimester data), but were not so during in vitro syncytialization of PHT cells ($P < 2.2 \times 10^{-16}$, K–S test, Fig. 5d). In fact, in the Azar et al.[34] data, which showed a mild 3′UTR lengthening (Fig. 5a), USPS genes were downregulated (Fig. 5d). Moreover, USPS gene expression levels were well correlated negatively with 3′UTR APA REDs in both the hESC model (Fig. 5e) and the BeWo cell model (Fig. 5f). Taken together, these data indicate that upregulation of genes involved in protein secretion functions, not cell syncytialization, is associated with 3′UTR shortening in SCT differentiation. We thus named this mechanism secretion-coupled APA (SCAP).

**3′UTR shortening is associated with increased RNA abundance.** We next wanted to understand the consequences of SCAP in SCT formation. We found that in both hESC and BeWo cell models, as well as in SCT far cells vs. VCT far cells in vivo, upregulated genes tended to show greater extent of 3′UTR shortening than genes without expression changes ($P < 1 \times 10^{-12}$, K–S test, Fig. 6a). Consistently, USPS genes in general displayed significant 3′UTR shortening in these conditions (Supplementary Fig. 8). On the other hand, downregulated genes displayed opposite trends ($P < 1 \times 10^{-13}$, K–S test, Fig. 6a). Note that we used RNA-seq reads in coding regions for gene expression analysis and those in 3′UTRs for APA analysis, avoiding confounding calculation (i.e., the same reads used for both expression and APA analyses). This result indicates that 3′UTR size regulation is related to gene expression changes. Consistently, the genes that showed shortened 3′UTRs in TB differentiation as defined by the 3′READS data (Fig. 2c) were significantly upregulated in both the hESC and BeWo cell models, as well as in SCT far cells vs. VCT far cells ($P < 1 \times 10^{-3}$, K–S test, Fig. 6b).

One way 3′UTRs could impact gene expression is through control of mRNA stability[40]. To explore this, we examined RNA stability of APA isoforms in BeWo cells. We metabolically labeled RNA with 4-thiouridine (4sU) for 1 h and compared the abundance of labeled RNAs, which represented newly made RNAs, with that of total cellular RNAs, which represented steady state RNAs (illustrated in Fig. 6c). As such, the ratio of abundance (steady state vs. newly made), named Stability Score, reflected transcript stability (Fig. 6c).

We found that long 3′UTR isoforms were generally less stable than short 3′UTR isoforms in BeWo cells by 9.4-fold (262 genes showing a long 3′UTR isoform being significantly less stable than a short isoform, and 28 genes showing the opposite trend, FDR = 0.05, DESeq, Fig. 6d). Importantly, as aUTR size increased, the stability difference between short and long 3′UTR isoforms became more evident (Fig. 6e). For example, for genes with an aUTR size >1836 nt (bin 5 in Fig. 6e), the median Stability Score difference was −0.61, whereas the difference was −0.16 for genes with an aUTR size <152 nt ($P < 2.2 \times 10^{-16}$ between these two groups test, Wilcoxon test, Fig. 6e). For *DNAJC3*, a USPS gene in bin 5, its long 3′UTR isoform was substantially less stable than its short 3′UTR isoform (Stability Scores = −1.7 vs. 1.5, Fig. 6f).

In contrast to 3′UTR shortening genes, the genes that displayed IPA activation showed only modest differences in gene expression as compared to all genes (Supplementary Fig. 9a). In addition, IPA isoforms were generally less stable than isoforms whose PASs were in the 3′ terminal exon (Supplementary Fig. 9b).

Using tetramer enrichment analysis, we found that several U-rich motifs were enriched in the aUTRs that were removed by APA in SCT differentiation (Supplementary Fig. 10a). Interestingly, these motifs were also associated with 3′UTRs of unstable transcripts in BeWo cells (Supplementary Fig. 10b). A general negative correlation could be discerned between tetramer enrichments in regulated aUTRs and in 3′UTRs for stability (Fig. 6g), indicating that removal of destabilizing motifs in aUTRs can potentially lead to increased expression for genes displaying 3′UTR shortening in SCT differentiation.

**SCAP in differentiation of B cells to plasma cells.** We next asked whether SCAP took place in other systems besides SCT differentiation. Differentiation of B cells to plasma cells was the first biological condition in which APA regulation was discovered[41,42]. Increasing protein secretion capacity from B cells to plasma cells is crucial for antibody secretion. We analyzed an RNA-seq dataset generated by Shi et al.[43], which included four types of B cells, namely, B1 cells, marginal zone B cells, germinal center (GC) B cells, and follicular B cells, as well as three types of plasma cells, namely, spleen plasmablasts (PBs), spleen plasma cells, and bone marrow plasma cells. Among these cell types, spleen PBs and marginal zone B cells displayed the shortest and longest 3′UTRs, respectively, based on 3′UTR APA REDs (Fig. 7a, *y*-axis). Significantly, 3′UTR sizes correlated well with USPS gene expression across these cell types ($r = -0.82$, Pearson correlation, Fig. 7a). By contrast, proliferation gene expression levels were not correlated with 3′UTR sizes ($r = -0.05$, Pearson correlation, Fig. 7b).

The difference between GC B cells and spleen PBs is quite striking, with the former having the highest proliferation gene expression levels (Fig. 7b, *x*-axis) and the latter having the highest USPS gene expression levels (Fig. 7a, axis). Importantly, consistent with the SCAP model, GC B cells had longer 3′UTRs than PBs. To corroborate this finding, we analyzed a dataset by Ise et al.[44], in which different types of GC B cells were isolated, including plasma cell fate B cells in the light zone, recycling cell fate B cells in light zone, and dark zone B cells. In addition, GC-derived PBs were used as a control. In agreement with the data by Shi et al.[43], PBs had shorter 3′UTRs than all GC B cells (Fig. 7c). Importantly, compared to dark zone B cells, which had the longest 3′UTRs among the three GC B-cell types (Fig. 7c), PBs had significantly higher expression of USPS genes and lower expression of cell proliferation genes ($P = 2.2 \times 10^{-12}$ and = $1.3 \times 10^{-6}$, respectively, K–S test, Fig. 7d). This result confirms that USPS gene expression is a good indicator of 3′UTR size in B-cell differentiation to plasma cells.

We next took an exploratory approach to identify genes whose expression changes were related to 3′UTR size differences across the B cells and plasma cells analyzed in the Shi et al.[43] study. Interestingly, using Pearson Correlation analysis (example cases shown in Supplementary Fig. 11a), we found an enrichment of genes with negative correlation coefficients (note the uneven data distribution in Supplementary Fig. 11b). GO analysis of the genes with the most negative correlation coefficients ($r < -0.79$, bottom 5% of all genes) revealed that these genes were highly enriched with functions in protein secretion, including "negative regulation of response to endoplasmic reticulum stress" and "protein exit from endoplasmic reticulum" (Supplementary Fig. 11c), further supporting the SCAP mechanism in B-cell differentiation.

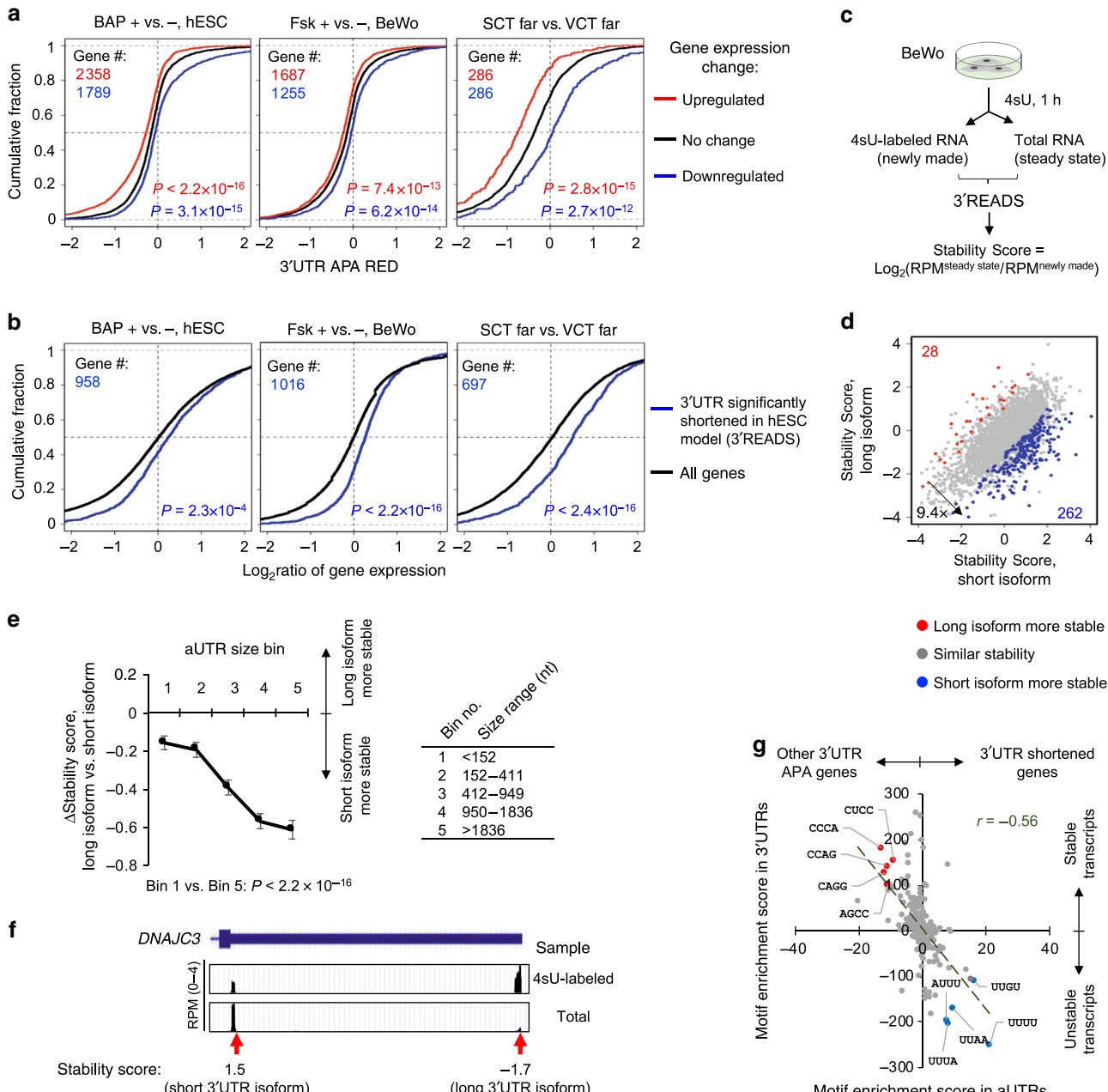

**Fig. 6 3′UTR shortening is associated with increased mRNA abundance. a** CDF curves of 3′UTR APA REDs for upregulated (red line), downregulated (blue line), and no change (black line) genes. Left, hESC model (>70 μm); middle, BeWo model (day 3); right, SCT far cells vs. VCT far cells (Vento-Tormo et al. and Tsang et al. data combined). Number of genes in each gene set is indicated. Regulated genes for the hESC and BeWo models are those with fold change > 1.2 and *P*-value < 0.05 (DESeq), and regulated genes in SCT far cells vs. VCT far cells are top and bottom 10% genes based on fold change. *P*-values (K–S test) for significance of difference between red or blue genes and black genes are indicated. **b** CDF curves of gene expression changes for genes showing 3′UTR shortening in the 3′READS data (Fig. 2c, blue line) and all genes (black line) in three datasets (same as in **a**). *P*-values are based on the K–S test comparing blue and black genes. **c** Schematic of mRNA stability analysis of BeWo cells. mRNA stability of each transcript is represented by a Stability Score, as indicated. **d** Scatter plot comparing Stability Score of short 3′UTR isoform (*x*-axis) with that of long 3′UTR isoform (*y*-axis). Genes whose isoforms have significantly different Stability Scores (*P* < 0.05, DEXSeq) are highlighted in color. **e** Relationship between aUTR size and Stability Score difference between long and short isoforms. Genes are grouped into five similarly sized bins (about 550 genes each) based on their aUTR length. *P*-value (Wilcoxon test) comparing Bins 1 and 5 is indicated. Data are presented as median value ± SEM. **f** UCSC Genome Browser tracks showing 3′READS data for *DNAJC3* in total and 4sU-labeled RNA samples. **g** Tetramer enrichment scores for aUTRs of genes with shortened 3′UTRs (Fig. 2c) vs. aUTRs of other genes (*x*-axis) are compared to enrichment scores for 3′UTRs of stable vs. unstable transcripts (*y*-axis).

**Perturbation of cell secretion further confirms SCAP.** We next wondered whether perturbation of cellular secretion capacity could lead to 3′UTR size changes. To this end, we analyzed an RNA-seq dataset involving overexpression (OE) of active forms of XBP1 (spliced form) and Creb3l2 (cleaved form) in the mouse pituitary tumor cell line AtT-20[45]. OEs of these factors individually or jointly led to enhancement of secretory capacity to various degrees[45]. As expected, Creb3l2 OE, which was reported to control expression of translation factors[45], had only a modest impact on USPS gene expression (median log$_2$Ratio = 0.03,

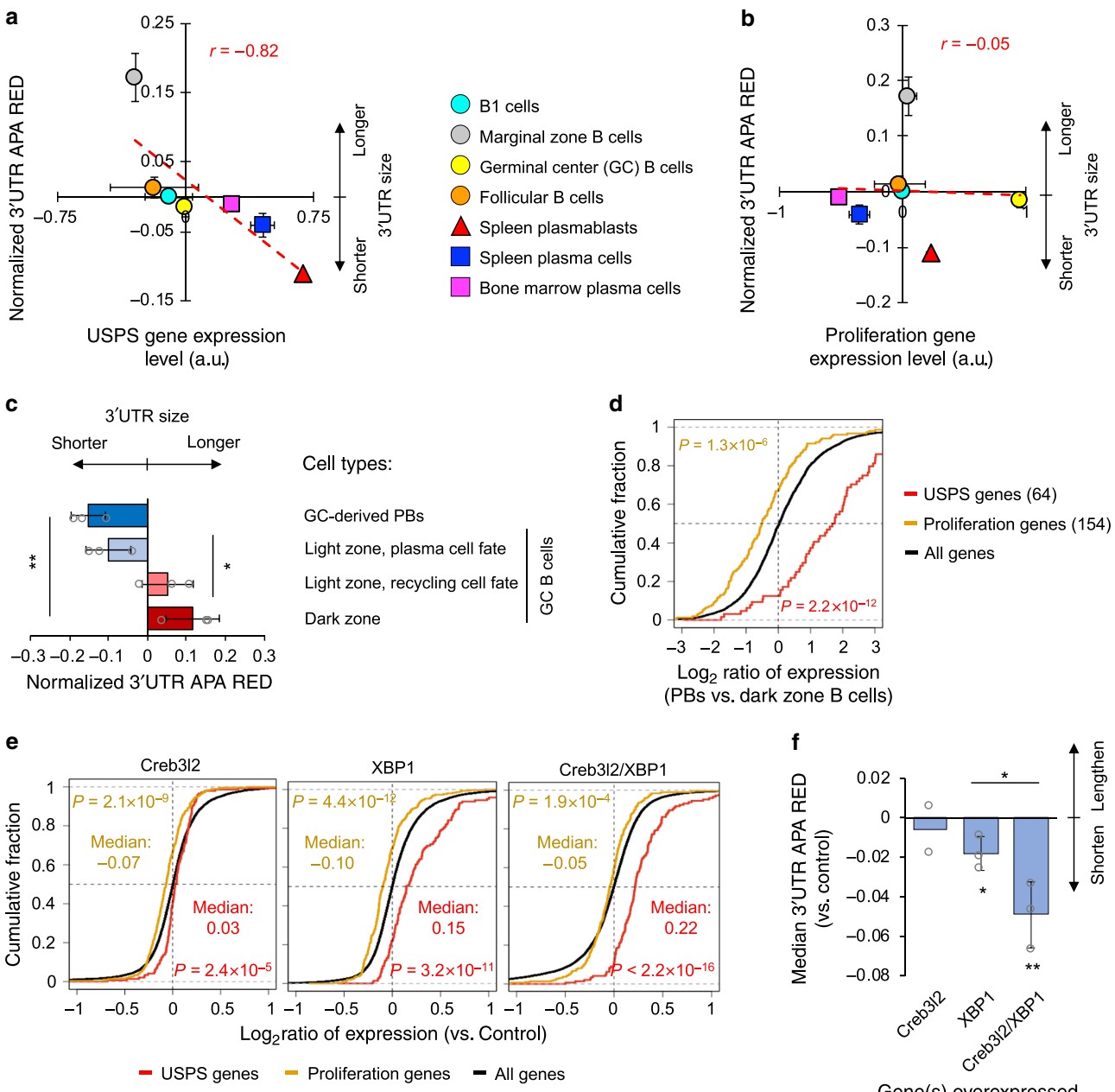

**Fig. 7 Additional systems supporting SCAP. a** Correlation between 3′UTR APA REDs and USPS gene expression levels across different types of B cells and plasma cells (a study by Shi et al.[43]). 3′UTR APA REDs are normalized to the median of all samples. Pearson correlation coefficient (*r*) is indicated. There are two samples for each cell type except for spleen plasmablasts (1 sample), spleen plasma cells (3 samples), and bone marrow plasma cells (1 sample). Data are presented as mean value+/− SD. a.u., arbitrary unit. **b** As in **a**, except that correlation between 3′UTR APA REDs and proliferation gene expression level is shown. **c** 3′UTR APA REDs for different types of B cells in the germinal center (GC) and GC-derived plasmablast (PB)(a study by Ise et al.[44]). 3′UTR APA REDs are normalized to the median of all samples. Data are presented as mean value+/− SD of three replicates. Significance of difference (*t*-test) between cell groups is indicated. Plasma cell fate B cells in the light zone are Bcl6[low] and CD69[high] cells, and recycling cell fate B cells are Bcl6[high] and CD69[high] cells. **d** CDF curves of gene expression changes for USPS (red) and proliferation (orange) genes in GC-derived PBs vs. dark zone B cells. Number of genes in each set is indicated. *P*-values (K–S test) for significance of difference between each gene set and all genes are indicated. **e** CDF curves of gene expression changes for USPS genes (red, 145), proliferation genes (orange, 298), and all genes in AtT-20 cells with overexpression of indicated gene(s) vs. control cells (a study by Khetchoumian et al.[45]). *P*-values (K–S test) for significance of difference between gene sets are indicated. **f** Bar plot of median 3′UTR APA REDs for the samples in **e**. Data are presented as mean value+/− SD based on two (Creb3l2) or three (XBP1 and Creb3l2/XBP1) replicates. Significance of difference (*t*-test) vs. control or between cell groups is indicated.

Fig. 7e, left). By contrast, XBP1 OE, which upregulates endoplasmic reticulum (ER) biogenesis[45], led to significant upregulation of USPS genes (median log₂Ratio = 0.15, Fig. 7e, middle). Interestingly, co-OE of both factors led to the greatest upregulation of USPS genes (median log₂Ratio = 0.22, Fig. 7e, right),

indicating synergy between these two factors. Consistent with the SCAP model, 3′UTR shortening was significant in cells with XBP1 OE and co-OE of XBP1 and Creb3l2 (Fig. 7f), with the latter being greater in magnitude than the former. By contrast, Creb3l2 OE did not elicit significant 3′UTR shortening (Fig. 7f).

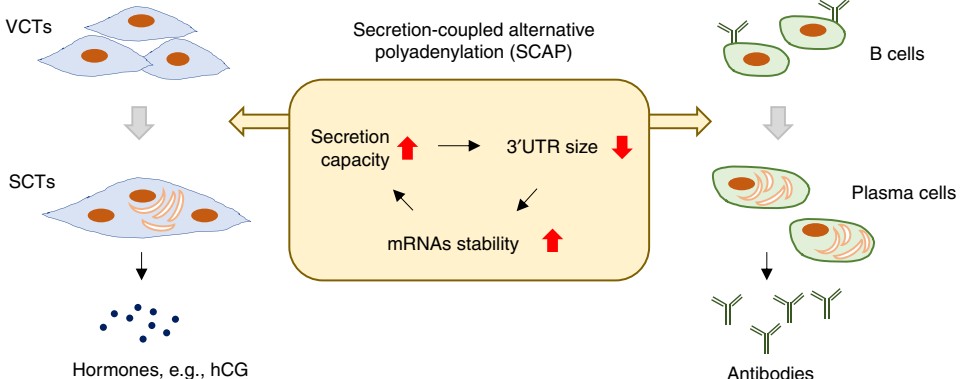

**Fig. 8 Secretion-coupled APA (SCAP) in differentiation of secretory cells.** SCAP mechanism in SCT differentiation and B-cell differentiation. IPA activation is not shown in the diagram.

Notable also is that proliferation genes were slightly down-regulated in all OE samples (orange lines in Fig. 7e), arguing against the possibility that the observed 3′UTR shortening was due to increased cell proliferation. Therefore, forced enhancement of secretion capacity through XBP1 or XBP1 + Creb3l2 OEs could also elicit 3′UTR shortening, lending a strong support to the SCAP mechanism.

## Discussion

In this study, we reveal secretion-coupled APA, or SCAP (illustrated in Fig. 8), involving global 3′UTR shortening and IPA activation in SCT differentiation. This mechanism is also in play during differentiation of B cells to plasma cells, indicating its generality in secretory cell formation. We further show that 3′UTR shortening is coupled with upregulation of gene expression, due likely to prolonged transcript half-life through removal of destabilizing sequences in 3′UTRs. We thus advocate that SCAP is an integral part of gene expression program in secretory cells, tailoring their transcriptome for enhanced protein production and secretion.

We show that SCAP is different from cell proliferation-elicited APA, which has been implicated in 3′UTR shortening in T-cell activation[20] and cancer cell development[46], or development-associated APA, which is executed in embryonic development and many cell differentiation lineages[21]. These conclusions are based largely on expression patterns of USPS genes, proliferation genes, and development genes. However, during the course of this study, we also noticed that gene expression levels of USPS genes and proliferation genes could be correlated in some systems. For example, in T-cell activation, where there is global 3′UTR shortening[20] (Supplementary Fig. 12a), both USPS genes and proliferation genes are activated (Supplementary Fig. 12b). A similar observation was made using data from a set of NCI-60 human tumor cell lines[47] (Supplementary Figs. 12c, d). Therefore, in most cells, secretion capacity (represented by USPS genes) appears to be coupled with cell proliferation. An open question that requires further investigation, therefore, is whether it is cell proliferation or protein secretion that is the primary driver for APA regulation in these non-secretory cells.

While 3′UTR shortening is widespread in secretory cell differentiation, it appears to impact secretion genes to a greater extent than other genes, such as those involved in ER stress response, ER organization, and secretory pathways. One ramification of SCAP, therefore, is amplification of transcriptional activation of secretion genes through stabilization of transcripts, leading to rapid accumulation of their RNAs in secretory cell differentiation. A case in point is *DNAJC3*, whose protein product (also known as P58^IPK) plays critical roles in mitigating ER stress,

including prevention of translational inhibition caused by stress-induced eIF2α phosphorylation[48] and function as a co-chaperone to protect stressed ER[49]. Mutations of *DNAJC3* have been implicated in diabetes and neurodegeneration[50]. Here, we show significant 3′UTR shortening of *DNAJC3* in SCT differentiation, and its two 3′UTR isoforms are conspicuously different in size and stability. It is thus conceivable that APA could augment *DNAJC3* protein expression to preemptively address potential ER stress when differentiating secretory cells ramp up their secretion capacity.

Another possible impact of 3′UTR shortening is alternation of mRNA localization, such as the association with ER[51] or other organelles[52]. This may lead to modulation of localized translation. The ER interaction may deserve special attention, given our observation that overexpression of XBP1, an activator of ER biogenesis, is sufficient to cause global 3′UTR shortening in the pituitary tumor cell line AtT-20. In this vein, it would be worthwhile to test how perturbation of APA might alter ER metabolism in secretory cells.

It is also worth noting that the genes encoding secreted proteins do not necessarily display APA. For example, many hormone-encoding genes, such as hCG genes (the major hormones secreted by SCTs), contain only one PAS in their 3′UTR (Supplementary Table 4). Therefore, 3′UTR shortening in general functions to facilitate the growth of secretory capacity in the cell rather than to help the production of individual secreted proteins.

Our result also suggests that some genes may employ APA to express protein isoforms through intronic PAS usage, which is globally activated during SCT differentiation. Notably, intronic PAS activation of the immunoglobulin M heavy chain gene in B-cell differentiation is one of the first reported APA events[41,42]. This regulation switches expression of a membrane-associated protein isoform in B cells to a secreted isoform in plasma cells. How widespread is protein isoform switch in SCT differentiation and in other secretory systems needs to be further delineated.

Global 3′UTR shortening could be caused by activated usage of proximal PASs during pre-mRNA processing (processing-driven) or by enhanced degradation of long 3′UTR isoforms (degradation-driven). We think the former is more likely in SCT differentiation because of two reasons. First, genes that display shortened 3′UTRs tend to have increased RNA abundance, making the processing-driven model more plausible. If the degradation-driven mechanism was the underlying cause, the RNA abundance increase could only be explained by a non-parsimonious scenario where enhanced RNA degradation is countered by transcriptional activation. Second, more importantly, we observed increased expression of IPA isoforms during SCT differentiation, which is in line with the notion that 3′ end

processing activity is activated. Note that IPA isoforms are typically less stable than isoforms using 3′ terminal exon PASs, further lending a support to the processing-driven model.

A growing number of factors have been implicated in global regulation of APA[12], including core cleavage and polyadenylation factors[53], U1 snRNP[54], transcriptional elongation[55], nuclear export[56], etc. How these mechanisms are in play in secretory cell differentiation is an open question. Notably, upregulation of CstF64, one of the core 3′ end processing factors, was previously found to be regulate APA in B-cell differentiation[57]. Whether different secretory cells share the same APA regulatory mechanism needs to be explored in the future.

## Methods

**Cell culture**. Human embryonic stem cells (hESCs) were differentiated into the trophectoderm lineage by using the BMP4 method[27]. BeWo choriocarcinoma cell line was a gift from the Sergei Kotenko lab at Rutgers University. BeWo cells were cultured in Dulbecco's modified Eagle's medium/F12 (DMEM/12) supplemented with 10% fetal bovine serum (FBS). Syncytialization of BeWo cells was induced by treating cells with 50 μM Forskolin (Fsk, Sigma-Aldrich) for 72 h. Mouse embryonic stem cells with doxycycline-inducible *Hras* mutant *Hras*$^{Q61L}$ were cultured on a tissue culture dish precoated with 0.5% gelatin in media containing leukemia inhibitory factor (LIF)[35]. At the time of induction with doxycycline (24 h after LIF removal), culture media was replaced with Dulbecco's modified Eagle's medium/F12 (DMEM/F12) containing 10% FBS. All culture media also contained 100 IU/ml penicillin and 100 μg/ml streptomycin.

**Immunofluorescence**. Cells grown on glass coverslips were fixed by using 4% paraformaldehyde in PBS for 10 min and permeabilized with PBS containing 0.1% Triton X-100 for 10 min. After blocking with 1% bovine serum albumin in PBST (PBS + 0.1% Tween 20) for 30 min, samples were immunostained with rabbit anti-e-cadherin (Cell Signaling, #3195, 1:200 in PBST) overnight. FITC-conjugated goat anti-rabbit antibody (Jackson ImmunoResearch, 111-095-144, 1:500 in PBST) was applied for 60 min. Cells were mounted on glass slides in SlowFade Gold Antifade reagent (Thermo Fisher) with DAPI. Fluorescence images were collected by using the EVOS FL Auto Cell Imaging System (Thermo Fisher).

**Real-time quantitative PCR (RT-qPCR)**. Total RNA was extracted by using TRIzol (Thermo Fisher) and treated with Turbo DNase (Thermo Fisher). Complementary DNA (cDNA) was synthesized by using oligo(dT) and M-MLV reverse transcriptase with 1.5–2 μg of RNA. cDNA was then mixed with real-time PCR primers and Luna qPCR master mix (NEB). qPCR was run on an Applied Biosciences StepOne Plus Real-Time PCR system.

**Isolation of newly made RNAs**. BeWo cells were cultured in medium supplemented with 50 μM of 4-thiouridine (4sU; Sigma) for 1 h before harvest. Total RNA (steady state) was extracted by using TRIzol. Newly made (4sU-labeled) RNAs were fractionated following the protocol described in ref. [58]. Briefly, 100 μg of total RNA was biotinylated by using biotin-HPDP (1 μg/μl in Dimethylformamide; Thermo Fisher Scientific), and then extracted with chloroform three times and precipitated with ethanol. The biotinylated RNA was captured by Streptavidin C1 Dynabeads. After washing for six times, biotinylated RNA was eluted by DTT. Both newly made and steady state RNAs were precipitated with ethanol and then used for 3′READS analysis.

**3′READS and data processing**. The 3′READS or its newer version 3′READS + procedures were performed to sequence the 3′ end fragment of poly(A) + RNAs[11,59]. Libraries were sequenced on an Illumina GAIIx (1 × 72 nt) or HiSeq (1 × 150 nt) platform. 3′READS/3′READS + data were then processed to identify polyadenylation site-containing reads, or PAS reads[59]. Briefly, the sequence corresponding to 5′ adaptor was first removed from raw reads by using Cutadapt[60]. Reads with short inserts (< 23 nucleotides) were discarded. The remaining reads were then mapped to the genome (hg19 for human or mm9 for mouse) by using bowtie2 (local mode)[61]. The 5′ random nucleotides derived from the 3′ adaptor were removed before mapping. Reads with a mapping quality score (MAPQ) ≥ 10 were kept for further analysis. Reads with ≥2 non-genomic 5′-Ts after alignment were called PAS reads. PASs within 24 nucleotides from each other were clustered[11]. The PAS read counts of genes were normalized by the median ratio method in the DESeq program[62].

**Analysis of 3′READS data**. For 3′UTR APA analysis, the two most abundant APA isoforms (based on PAS reads) whose PASs are in the 3′UTR of the last exon were selected. They were named proximal PAS (pPAS) and distal PAS (dPAS) isoforms. Significant APA events were those with relative abundance change > 5% and *P*-value < 0.05 (Fisher's exact test or DEXSeq analysis) between samples. Relative

Expression Difference (RED) was calculated as the difference in log2(RPM ratio) of the two APA isoforms between two samples. The aUTR size was the distance between the two APA sites in the 3′UTR. For Intronic APA analysis, both PAS reads in intronic regions (IPA isoforms) and PAS reads in the last exon (TPA isoforms) were respectively summed and compared. Significant APA events were those with relative abundance change > 5% and *P*-value < 0.05 (Fisher's exact test or DEXSeq analysis) between samples.

**Analysis of APA using RNA-seq data**. 3′UTR APA analysis using RNA-seq data were carried out by the SAAP-RS method[25]. Briefly, reads were aligned to the human genome with STAR v2.5.2[63] by using default settings. Raw bam files were further processed by using the R packages RSamtools (for processing bam files), GenomicAlignments (for counting reads), and GenomicFeatures (for defining genomic regions). Human PAS locations were obtained from PolyA_DB3[64]. Relative expression difference (RED) was calculated as difference in log2(aUTR read number/cUTR read number) between two samples. Significance of difference was analyzed by either DEXSeq when there were replicates or the Fisher's exact test when there were no replicates. When there were no replicates, standard deviation was obtained by sampling data with a bootstrapping method for 20 times[53].

**Gene sets used in this study**. For the TB subtype marker gene panel, four trophoblast studies were used[28–31]. Each marker gene for a specific TB subtype was reported by at least two studies and had no conflict between studies. Proliferation gene set was based on the study by Sandberg et al.[20]. Development gene set contained the genes whose expression levels positively correlated with 3′UTR size in mouse embryonic development, as defined in the study by Ji et al.[21] (called PCS in that study). The USPS gene set defined in this study contained genes whose expression levels increased in both hESC and BeWo models and were associated with the top three GO terms for upregulated genes in the two models.

**Motif analysis**. The frequencies of tetramers in specific regions were calculated and compared between different gene sets. Motif enrichment scores were calculated by –log10(*P*-values), where *P*-values were based on the Fisher's exact test.

**Gene expression analysis using single-cell RNA-seq data**. Droplet-based single-cell sequencing data[29,30] were processed by using the Cell Ranger Single-Cell Software Suite (v3.0.1, 10x Genomics). SMART-Seq2 data were processed as described by Tsang et al.[36]. Briefly, raw reads were assigned to cells based on cell-specific barcodes in read 2 of paired-end reads. Read 1 reads were trimmed by using CutAdapt[60] to remove the template switch oligo sequence and the poly(A) tail sequence. All reads were aligned to hg19 genome by using STAR (v2.5.2)[63].

**Single-cell significance analysis of alternative polyadenylation (scSAAP)**. Single-cell reads were mapped to the hg19 genome by using STAR[63]. Single cells were clustered with the Seurat package[65]. TB clusters were extracted and subtyped based on the TB marker gene panel. Reads from single cells in each cell cluster were summed, and each cell cluster was treated as a bulk RNA-seq sample for 3′UTR APA analysis. Pseudotime analysis was performed by using the monocle package[37]. The TB marker panel was used to order single cells along the TB subtype branches. The cells in each TB subtype branch were then divided equally in cell number into two groups ("near" and "far" based on distance from the center).

**Gene ontology analysis**. Gene ontology (GO) analysis was carried out by using the GOstats Bioconductor package[66], where the hypergeometric test was performed without multiple comparison adjustment. Generic terms (associated with >1000 genes) were discarded. To reduce redundancy in reporting, any GO term with a greater than 75% gene overlap with a more significant term was discarded.

**Statistics and reproducibility**. The Student's *t*-test was used to determine statistical significance between groups; all tests are two-sided. The Fisher's exact test or DEXSeq were used to determine the significance of APA changes. The K–S test was used to compare distributions between gene sets. The Wilcoxon test was used to compare gene expression changes in different gene sets; all tests are two-sided. Boxplots were generated with the top bound defined as the third quartile, the bottom bound defined as the first quartile, and the center defined as the median. Where shown, the following symbols indicate statistical significance: n.s., $P \geq 0.05$; *$P < 0.05$; **$P < 0.01$, ***$P < 0.001$.

**Reporting summary**. Further information on research design is available in the Nature Research Reporting Summary linked to this article.

## Data availability

Sequencing datasets that were generated in this study have been deposited into the NCBI GEO database under the accession number GSE138759. All datasets used in this study are listed in Supplementary Table 5. All data is available from the corresponding author upon reasonable request.

## Code availability

All custom-made code and scripts for processing of sequencing data and quantitative analyses were written in Perl or R, and will be provided upon request.

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

## Acknowledgements

We thank members of BT lab for helpful discussions, Nick Illsley for technical advice on BeWo cells, and contributions from Wencheng Li at an early stage of this work. This work was funded by NIH grants: T32 GM008339 to L.C.C., R21 HD081682 to B.T. and C.W.L., R35 GM118136 to J.L.M., and R01 GM084089 and R01 GM129069 to B.T.

## Author contributions

B.T. conceived and designed experiments. L.C.C. and B.T. conceived and designed data analysis. D.Z., E.B., K.O., M.H. and P.L.Y. performed experiments. L.C.C. and F.S. performed data analysis. C.W.L. and J.L.M. participated in experiments. L.C.C. and B.T. wrote the paper.

## Competing interests

The authors declare no competing interests.
