## [Peer Review File · Nature Communications]

Reviewers' comments:

Reviewer #1 (Remarks to the Author):

In this generally well-written paper, Cheng et al. observed a widespread transcript shortening during syncytiotrophoblast differentiation, which the authors link to polyadenylation at more proximal polyA sites, which, in turn, they propose leads to enhanced protein secretory activity, most probably as the result of increased transcript half-life in the secretory cell. The group use data acquired from a number of models for trophoblast differentiation, including BMP4-mediated differentiation of human embryonic stem cells (ESC) and conversion of several transformed human trophoblast cell lines to more differentiated cell types. They have also re-analyzed extensive amounts of RNAseq information on trophoblast cell types (some of it single cell) available through publicly accessible data bases. Importantly the group demonstrates that the differentiation of B-cells to plasma cells actively secreting immunoglobulin also results in a similar 3'/UTR shortening of transcripts. In other words, the phenomenon is not confined to trophoblast, but probably to other cases where there is differentiation of an early lineage precursor to a highly secretory cell type, a fact that is not altogether made clear from the title

Although this reviewer is persuaded that the phenomenon of UTR shortening of transcripts in secretory trophoblast is real, there are a number of issues that should be addressed.

1. The paper is difficult to read because of the number of acronyms, most of which are non-standard and certainly unfamiliar to this reviewer. Either provide a reference table or preferably reduce the list to those terms used most frequently.
2. There is no information provided either in the paper itself or in the cited reference (27) for how ESC are driven to trophoblast by the BMP protocol. This issue is important because the concentration of BMP4, the length of exposure, the gas atmosphere, and even the quality of the BMP4 itself influences outcomes (Ref. 7). It is particularly important that FGF2 signaling be minimized because the presence of this growth factor plus the BMP4 results in emergence of mesoderm and endoderm, as well as trophectoderm, depending upon the relative concentrations of the two growth factors.
3. The authors are incorrect in believing that the HTR8 cell line is derived from a choriocarcinoma. In fact, it is an extravillous trophoblast (EVT) cell line that had been transformed with SV40. Accordingly, I am not sure whether the JEG3 vs. HTR8 comparisons (Fig. 5) are appropriate for the hypothesis under examination.
4. The authors showed that there was 3'/UTR shortening of seven genes (p 6, bottom of first paragraph), although it is not clear whether others were similarly analyzed (the choice is modified by the term "including"). This is a curious choice for an attempt to prove consistency with their hypothesis that genes encoding secretory proteins should be subject to increase 3'/UTR shortening. First, I was unable to identify a gene with the name PRDX6, while, of the other five, only TIMP2 encodes a proven secretory protein. Also, TIMP2 is expressed only modestly in syncytiotrophoblast relative to other genes upregulated during differentiation, e.g. CGA (FPKM 40.8 vs 64,929). Also, according to the data of Yabe et al., SCD is more strongly expressed in the progenitor ESC than in the differentiated syncytiotrophoblast derived from them. It is suggested that the authors reanalyze their choice of genes and select them in an unbiased manner from ones that clearly encode secretory products and that are

highly upregulated in syncytiotrophoblast relative to the precursor cell.

Minor issues:

1. The final paragraph of the Introduction is redundant. It merely repeats what is stated in the Abstract.
2. The use of the participle “using” is tricky if there is constant switching between the active and passive voice. For example it is incorrect to state “Cells on coverslips were fixed using paraformaldehyde..” because the cells didn’t use anything; you used the formaldehyde. Therefore “using” is said to be dangling. Putting the participle “by” before “using” corrects the grammatical error. On the other hand, it is perfectly fine to state that “We fixed the cells using paraformaldehyde.” There are numerous examples throughout Materials and Methods where this error occurs, and it is suggested that the errors be corrected.

Reviewer #2 (Remarks to the Author):

In this paper, Bin Tian et al. report that unlike many other cell differentiation lineages, differentiation of syncytiotrophoblasts (SCTs), elicits widespread transcript shortening through APA in 3’UTRs and in introns. The shortening event is unrelated to cell proliferation, but instead accompanies increased cellular function in protein secretion. This is a novel finding and provides a strong evidence that 3’UTR shortening and IPA isoform upregulation are critical for secretory cells function. The strengths of the paper are that this observation appears quite novel and the authors employ a battery of robust computational approaches to characterize the observation in multiple experimental systems. The weaknesses include the concern that the work is largely descriptive making it difficult to assess how significant the SCAP phenomenon is. There are a few concerns with the work presented in the paper, as listed below.

1. In Fig. 1b and c, it is not clear why RNA-seq reads in 3’UTRs are divided into cUTR and aUTR groups based on the first and last PASs. In normal cells, common UTR should be a canonical UTR with a distal poly(A) site instead of the first PAS, and aUTR should be a non-canonical UTR with shorten or extend UTR.
2. Fig. 2g, UCSC genome browser tracks for DNAJC3 in BeWo syncytialization didn’t show APA changes between Fsk+ and -, but Fig 2h. showed DNAJC3 was shortened with Fsk+ treatment.
3. Integrative analysis connects 3’UTR size regulation to protein secretion, it is interesting to know whether these SCAP genes in Fig. 4b overlap with B cell markers or the listed genes in Fig 2f?
4. Fig 1d, I understand the authors argument but I feel that expression of a proliferation genes as a proxy for actual proliferation measurements may have more caveats than the authors describe

5. Figure 1f. The way the authors wording is 'we found that the longer the aUTR, the greater the 3'UTR shortening' to be circular. Of course this is true. Perhaps the sentence needs some additional wording to clarify: the longer the aUTR, the higher usage of proximal to distal PAS was observed? Or something to this nature?

6. In figure 2, I am concerned about the use of the BeWo model and the robustness of the data presented. The 3READS data for the hESC model is very clear for a gene like DNAJC3 but the RNAseq data for the retrospective analysis of the FSK-induced BeWo cell fusion model is not very convincing at best and concerning at worst. Moreover, it appears that the authors have repeated this FSK-induced fusion as they show validation data making this reviewer desire 3READS of this experiment as the authors conducted it (rather than retrospective analysis).

7. It is confusing why the sc-RNA-seq data in figure 3d appears the way it does. There are distinct peaks above each of the two PASs yet this is not 3READS?? I understand that the authors are analyzing previous data but one would expect RNA-seq to have more equal coverage. Is it concerning then to apply SAAP-RS to read coverage like this?

8. The significance cutoffs for the DESeq described in figure 4 seem very relaxed. The data would be more compelling if fold change was at least 1.5 fold (preferably 2) with a p-adj <0.01.

9. The authors present a correlation argument that JEG3 cells secrete better than HTR8 cells and have greater degree of 3'UTR shortening and IPA but it is not clear if this is responsible for the increased secretion in this system. So at this point, the argument is correlative only. If the authors can provide some evidence that if they: 1) cause broad lengthening of 3'UTRs in JEG3 cells that secretion is reduced or 2) cause broad 3'UTR shortening in HTR8 cells is sufficient to increase secretion. Tools exist to attempt these experiments.

10. Along these same lines, can the authors determine if JEG3 cells enhanced secretion is restricted to an exogenous luciferase reporter or includes several endogenous secreted proteins?

11. The authors show an enrichment in U-rich motifs near PASs subject to 3'UTR shortening during SCT differentiation suggesting that the U-rich motif is a driving factor governing this phenomenon. To support this model, they use a luc reporter harboring a TIMP2 PAS. I had trouble understanding the rationale for choosing this particular TIMP2 PAS. Does this particular PAS have a nearby U-rich motif? I assumed so but the authors don't state this. If it does, then I recommend an additional control with that U-motif being mutated.

Reviewer #3 (Remarks to the Author):

This study by L. Cheng et al. shows a very interesting link between alternative polyadenylation (APA) and trophoblast (TB) differentiation. APA emerges as an important layer of gene regulation as the majority of protein coding genes contain multiple poly(A) sites in their 3'UTR. Yet, this regulatory layer is largely unexplored and our understanding of its biological roles is only rudimentary. The main novelty and significance of this manuscript lies in this point: while global 3'UTR lengthening and shortening were observed before under many biological processes, the biological impact of these changes usually remained elusive. In this study however, the modulation of APA was strongly linked with a definite biological endpoint: the genes experiencing 3'UTR shortening during differentiation of syncytiotrophoblasts (SCT) were significantly enriched for genes that function in protein secretion processes, and this change in 3'UTR length was significantly associated with increase in the expression level of these genes. This coupling was termed by the authors as secretion-coupled APA (SCAP). Last, the authors observed a similar trend in the maturation of B cells to plasma cells, that are characterized too by enhanced protein secretion, suggesting that SCAP may be a general regulatory scheme in cells with enhanced protein- secretion activity.

The analyses are well performed and most findings are supported by several datasets using different cellular models for the probed biological process. I have few suggestions/comments:

1. 3'UTR shortening can stem from either enhanced CPA activity or enhanced degradation of the longer isoforms (or both). The authors use a very nice system to show enhanced CPA in JEG3 vs. HTR8 (Fig. 6g). The potential contribution for enhanced destabilization of the longer isoforms during SCT differentiation was not examined at all. I think it will be important to analyze whether such effect contributes too to the observed phenomenon (similar to the global 3'UTR shortening that occurs during spermatogenesis). Could the authors examine for few relevant genes (preferably from the USPS set), whether the stability of the longer isoform is modulated during the differentiation process (using either cellular model used in the paper for this process)? (relevant to this point, the authors mention that "Notably, genes that showed significant 3'UTR shortening had enriched U-rich motifs in their aUTRs (Fig S8a)" – suggesting involvement of destabilization of the longer isoform during SCT differentiation.

2. Page 6, 3rd line: Fig. S3b is argued to present the consistency of APA regulation between the hESC and BeWo models. However, the effect shown in this Fig is weak. Could the authors also present a scatter plot with RED values measured in these two models. This will provide a better way to assess the overall consistency.

3. (a) For all Figs showing RED scores (main and supplementary), indicate in the Y-axis label (or in the legend) what was the reference level (for example, this is clearly indicated in Fig 2f but not 2d). (b) For all plots (main and supplementary) showing CDF (e.g., Fig3c), please add an indication for the number of genes in each gene set.

4. Fig 4e-f show calculation of r based on only 4 data points. This measure is very sensitive for such a small n (adding one additional point is likely to drastically change the value of r), and therefore the indicated value is not really meaningful. I think that simply showing the plot for the observed tendency without specifying r is better. Same for Fig S12 that is based on only 3 points. Again, r in such case is not

really meaningful.

5. Could the authors color the USPS genes in a different color in Fig 5?

6. p11 – “we found an enrichment of genes whose expression levels were negatively correlated with 3’UTR size”. “enrichment” by which statistical test? Pval?

7. Fig S13. Why these specific genes are shown? Please explain in a legend.

Point-by-point Response

Reviewer #1

In this generally well-written paper, Cheng et al. observed a widespread transcript shortening during syncytiotrophoblast differentiation, which the authors link to polyadenylation at more proximal polyA sites, which, in turn, they propose leads to enhanced protein secretory activity, most probably as the result of increased transcript half-life in the secretory cell. The group use data acquired from a number of models for trophoblast differentiation, including BMP4-mediated differentiation of human embryonic stem cells (ESC) and conversion of several transformed human trophoblast cell lines to more differentiated cell types. They have also re-analyzed extensive amounts of RNAseq information on trophoblast cell types (some of it single cell) available through publicly accessible data bases. Importantly the group demonstrates that the differentiation of B-cells to plasma cells actively secreting immunoglobulin also results in a similar 3'/UTR shortening of transcripts. In other words, the phenomenon is not confined to trophoblast, but probably to other cases where there is differentiation of an early lineage precursor to a highly secretory cell type, a fact that is not altogether made clear from the title.

Thanks for the enthusiasm about our work. Thanks for pointing out that our title is not informative enough. We have modified the title accordingly.

1. The paper is difficult to read because of the number of acronyms, most of which are non-standard and certainly unfamiliar to this reviewer. Either provide a reference table or preferably reduce the list to those terms used most frequently.

We apologize for extensive use of acronyms. We have substantially reduced acronym use in our revision.

2. There is no information provided either in the paper itself or in the cited reference (27) for how ESC are driven to trophoblast by the BMP protocol. This issue is important because the concentration of BMP4, the length of exposure, the gas atmosphere, and even the quality of the BMP4 itself influences outcomes (Ref. 7). It is particularly important that FGF2 signaling be minimized because the presence of this growth factor plus the BMP4 results in emergence of mesoderm and endoderm, as well as trophectoderm, depending upon the relative concentrations of the two growth factors.

We apologize for this miscite. We have now cited the right publication (Xu et al. Nat Biotechnol. 2002), which included the detailed protocol. We appreciate the insights this reviewer provides about hESC differentiation. We understand that the method used by Yabe et al. might be the best in the field, as it contained an FGF2 inhibitor. This is why we focused on this data set for most of our analysis. However, with respect to gene expression changes, the other two hESC model data sets correlated well with the Yabe et al. data, which is now explicitly shown in Fig. S2b. Therefore, we do not think our conclusions would be altered by the protocol differences between the three studies.

3. The authors are incorrect in believing that the HTR8 cell line is derived from a choriocarcinoma. In fact, it is an extravillous trophoblast (EVT) cell line that had been transformed with SV40. Accordingly, I am

not sure whether the JEG3 vs. HTR8 comparisons (Fig. 5) are appropriate for the hypothesis under examination.

Thanks for this insightful comment. We understand that while the JEG3 vs. HTR8 comparison result is in line with our hypothesis, these two cell types may require additional characterization to be fully useful. We have thus decided to remove this part of the work.

4. The authors showed that there was 3'UTR shortening of seven genes (p 6, bottom of first paragraph), although it is not clear whether others were similarly analyzed (the choice is modified by the term "including"). This is a curious choice for an attempt to prove consistency with their hypothesis that genes encoding secretory proteins should be subject to increase 3'UTR shortening. First, I was unable to identify a gene with the name PRDX6, while, of the other five, only TIMP2 encodes a proven secretory protein. Also, TIMP2 is expressed only modestly in syncytiotrophoblast relative to other genes upregulated during differentiation, e.g. CGA (FPKM 40.8 vs 64,929). Also, according to the data of Yabe et al., SCD is more strongly expressed in the progenitor ESC than in the differentiated syncytiotrophoblast derived from them. It is suggested that the authors reanalyze their choice of genes and select them in an unbiased manner from ones that clearly encode secretory products and that are highly upregulated in syncytiotrophoblast relative to the precursor cell.

We thank the Reviewer for raising this important point. First, our intention for Fig. 2h was to confirm our RNA-seq analysis result. We thus selected a few genes based exclusively on the degree of 3'UTR shortening and their statistical significance. We did not specifically select USPS genes. We have now modified text to make this point more clear. Second, we now specifically present 3'UTR shortening of USPS genes (Fig. S8) to address this issue. As expected, USPS genes tend to display a greater degree of 3'UTR shortening as compared to other genes.

Minor issues:

1. The final paragraph of the Introduction is redundant. It merely repeats what is stated in the Abstract.

Thanks for the careful reading. We have fixed this issue.

2. The use of the participle "using" is tricky if there is constant switching between the active and passive voice. For example it is incorrect to state "Cells on coverslips were fixed using paraformaldehyde.." because the cells didn't use anything; you used the formaldehyde. Therefore "using" is said to be dangling. Putting the participle "by" before "using" corrects the grammatical error. On the other hand, it is perfectly fine to state that "We fixed the cells using paraformaldehyde." There are numerous examples throughout Materials and Methods where this error occurs, and it is suggested that the errors be corrected.

Thanks for the careful reading. We have fixed this issue.

Reviewer #2

In this paper, Bin Tian et al. report that unlike many other cell differentiation lineages, differentiation of syncytiotrophoblasts (SCTs), elicits widespread transcript shortening through APA in 3'UTRs and in introns. The shortening event is unrelated to cell proliferation, but instead accompanies increased cellular function in protein secretion. This is a novel finding and provides a strong evidence that 3'UTR shortening and IPA isoform upregulation are critical for secretory cells function. The strengths of the paper are that this observation appears quite novel and the authors employ a battery of robust computational approaches to characterize the observation in multiple experimental systems. The weaknesses include the concern that the work is largely descriptive making it difficult to assess how significant the SCAP phenomenon is.

We appreciate the general positive sentiment expressed by this reviewer, especially on the novelty and robustness of our work. We do acknowledge that many mechanistic details of our finding remain unresolved, which we intend to follow up on in the future. We do believe, however, that while the work is largely descriptive, it is highly original and its significance is readily palpable. This is also echoed by the two other reviewers. In the revision, we have further strengthened our SCAP model, including using additional B cell/plasma cell data and cells with overexpression of XBP1 and Creb3l2. Moreover, we have examined and discussed the relationship between SCAP and proliferation-related APA.

1. In Fig. 1b and c, it is not clear why RNA-seq reads in 3'UTRs are divided into cUTR and aUTR groups based on the first and last PASs. In normal cells, common UTR should be a canonical UTR with a distal poly(A) site instead of the first PAS, and aUTR should be a non-canonical UTR with shorten or extend UTR.

We defined common UTR (cUTR) as the region of 3'UTR shared by all 3'UTR isoforms while the alternative UTR (aUTR) is the region that is subject to APA regulation. Therefore, the 'common' used here does refer to expression level. We now use 'constitutive' instead for cUTR, to dispel any potential confusion.

2. Fig. 2g, UCSC genome browser tracks for DNAJC3 in BeWo syncytialization didn't show APA changes between Fsk+ and -, but Fig 2h. showed DNAJC3 was shortened with Fsk+ treatment.

We agree that 3'UTR shortening of *DNAJC3* in the BeWo model appears less substantial than in the hESC model, despite the fact that both are statistically significant. We now show both RED and P-value to make it easier to comprehend this result. It is also noteworthy that the overall extents of 3'UTR shortening in these two models are comparable, as shown in Figs. 2g and S3b.

3. Integrative analysis connects 3'UTR size regulation to protein secretion, it is interesting to know whether these SCAP genes in Fig. 4b overlap with B cell markers or the listed genes in Fig 2f?

If we understand the Reviewer's comment correctly, this reviewer wants to see how USPS genes are regulated in B cell differentiation. This data is shown in Fig. 7a (now Fig. 6a). We have also included another B cell data set in the revision (shown in Figs. 6c and 6d). In both cases, USPS correlated well with 3'UTR shortening.

4. Fig 1d, I understand the authors argument but I feel that expression of a proliferation genes as a proxy for actual proliferation measurements may have more caveats than the authors describe

Thanks for pointing this out. In view of this comment, we have included another gene set, which we previously identified to correlate with 3'UTR size in embryonic development (Ji et al. PNAS 2009). We name this as development gene set. In addition, we have used C2C12 cell differentiation as a reference for comparison with TB differentiation. All these analyses support the notion that APA in TB differentiation is different than previously identified proliferation- and development-associated APA.

5. Figure 1f. The way the authors wording is 'we found that the longer the aUTR, the greater the 3'UTR shortening' to be circular. Of course this is true. Perhaps the sentence needs some additional wording to clarify: the longer the aUTR, the higher usage of proximal to distal PAS was observed? Or something to this nature?

Sorry for the confusion. We meant the extent of shortening, not the physical size. We have now made it more clear.

6. In figure 2, I am concerned about the use of the BeWo model and the robustness of the data presented. The 3READS data for the hESC model is very clear for a gene like DNAJC3 but the RNAseq data for the retrospective analysis of the FSK-induced BeWo cell fusion model is not very convincing at best and concerning at worst. Moreover, it appears that the authors have repeated this FSK-induced fusion as they show validation data making this reviewer desire 3READS of this experiment as the authors conducted it (rather than retrospective analysis).

As noted above, the BeWo model is similar in extent of 3'UTR shortening to the hESC model. We now show both a scatterplot comparing these models (Fig. 2g), and histograms of 3'UTR APA REDs of the two models (Fig. S3b).

7. It is confusing why the sc-RNA-seq data in figure 3d appears they way it does. There are distinct peaks above each of the two PASs yet this is not 3READS?? I understand that the authors are analyzing previous data but one would expect RNA-seq to have more equal coverage. Is it concerning then to apply SAAP-RS to read coverage like this?

We thank the Reviewer for this thoughtful comment. The scRNA-seq data was generated by using the 10X Genomics platform, which gives rise to reads biased to the 3' end. Applying SAAP-RS would not affect the calculation as distal PAS isoform reads would still fall within the aUTR region. We now show *DNAJC3* data from another study by Tsang et al. (Fig. S5c), which shows a similar pattern.

8. The significance cutoffs for the DESeq described in figure 4 seem very relaxed. The data would be more compelling if fold change was at least 1.5 fold (preferably 2) with a p-adj <0.01.

We understand this reviewer's concern. Normally, one may want to use a more stringent cutoff to select genes. However, in the current context, where there is much variability between the experimental models, it may be desirable to use slightly less stringent cutoffs. In addition, since the final goal is to identify significant GO terms, too stringent cutoffs would result in much fewer genes, making the GO

analysis less sensitive. It is also notable that we used replicates in each model to ensure data robustness. Nevertheless, we have tried the 2-fold and FDR < 1% cutoff. As shown below, the result is similar.

Response Figure 1. a. Significantly regulated genes in hESC and BeWo models, as in Fig. 4b. Significant regulation is based on fold change > 2.0 and FDR = 1% (DESeq) for both models. Commonly regulated genes are highlighted. Red for upregulation and blue for downregulation. **b.** P-values for the top GO terms shown in Fig. 4c. The order of GO terms is the same as that in Fig. 4c. P-values are based on gene selection in a.

9. The authors present a correlation argument that JEG3 cells secrete better than HTR8 cells and have greater degree of 3'UTR shortening and IPA but it is not clear if this is responsible for the increased secretion in this system. So at this point, the argument is correlative only. If the authors can provide some evidence that if they: 1) cause broad lengthening of 3'UTRs in JEG3 cells that secretion is reduced or 2) cause broad 3'UTR shortening in HTR8 cells is sufficient to increase secretion. Tools exist to attempt these experiments.

We thank this reviewer for this great thought. First, in consideration of reviewer #1's suggestion, we have removed the JEG3 and HTR8 data. Second, we understand it would be quite exciting if perturbation of 3'UTR size through some known factors could lead to secretion changes. The result would greatly support our model. However, we think this kind of experiment might not that straightforward, for two major reasons:

a) A growing number of APA mechanisms have been reported recently (some are reviewed in Tian and Manley, Nature Review in Molecular and Cellular Biology, 2017). Just perturbing some factors to artificially change 3'UTR sizes might not be physiologically relevant. As our lab and others have shown, while different perturbations may lead to similar global trends, such as 3'UTR shortening or lengthening, different gene sets could be impacted on in different conditions. Therefore, the existing tools that this reviewer suggested may not be all that valuable in this context.

b) It is entirely unclear that at what stage of secretion pathway the 3'UTR shortening may play a role and whether the shortening needs to be coupled with other mechanisms to take effect. A case in point is a new data set we now present in the revision. While overexpression of *Creb3l2* has little effect on 3'UTR

shortening, it has a synergistic effect when *XBP1* is also overexpressed (Figs. 6e and 6f). Note that both *Creb3l2* and *XBP1* overexpression changes secretion capacity to some degree. Therefore, it is premature to posit that a simple perturbation of 3'UTR length would change secretion.

With the above said, we believe both aspects need to be studied in a more thorough fashion in the future. As such, we have now removed all the data related to polyA site regulation.

10. Along these same lines, can the authors determine if JEG3 cells enhanced secretion is restricted to an exogenous luciferase reporter or includes several endogenous secreted proteins?

Greater secretion of endogenous proteins have been documented before, e.g., Poloski et al. Biol Reprod, 2016. However, we have removed this data per reviewer #1's suggestion.

11. The authors show an enrichment in U-rich motifs near PASs subject to 3'UTR shortening during SCT differentiation suggesting that the U-rich motif is a driving factor governing this phenomenon. To support this model, they use a luc reporter harboring a TIMP2 PAS. I had trouble understanding the rationale for choosing this particular TIMP2 PAS. Does this particular PAS have a nearby U-rich motif? I assumed so but the authors don't state this. If it does, then I recommend an additional control with that U-motif being mutated.

We chose TIMP2 PAS reporter because it is one of the significantly regulated PASs in our sequencing analysis. We agree that we should carry out more rigorous experimentation to study the underlying mechanism. However, as noted above, we think this is beyond the scope of this paper. We have thus removed all data related to polyA site regulation.

Reviewer #3 (Remarks to the Author):

This study by L. Cheng et al. shows a very interesting link between alternative polyadenylation (APA) and trophoblast (TB) differentiation. APA emerges as an important layer of gene regulation as the majority of protein coding genes contain multiple poly(A) sites in their 3'UTR. Yet, this regulatory layer is largely unexplored and our understanding of its biological roles is only rudimentary. The main novelty and significance of this manuscript lies in this point: while global 3'UTR lengthening and shortening were observed before under many biological processes, the biological impact of these changes usually remained elusive. In this study however, the modulation of APA was strongly linked with a definite biological endpoint: the genes experiencing 3'UTR shortening during differentiation of syncytiotrophoblasts (SCT) were significantly enriched for genes that function in protein secretion processes, and this change in 3'UTR length was significantly associated with increase in the expression level of these genes. This coupling was termed by the authors as secretion-coupled APA (SCAP). Last, the authors observed a similar trend in the maturation of B cells to plasma cells, that are characterized too by enhanced protein secretion, suggesting that SCAP may be a general regulatory scheme in cells with enhanced protein- secretion activity. The analyses are well performed and most findings are supported by several datasets using different cellular models for the probed biological process.

We thank this reviewer for high enthusiasm about our work, and precise understanding of the novelty and significance.

1. 3'UTR shortening can stem from either enhanced CPA activity or enhanced degradation of the longer isoforms (or both). The authors use a very nice system to show enhanced CPA in JEG3 vs. HTR8 (Fig. 6g). The potential contribution for enhanced destabilization of the longer isoforms during SCT differentiation was not examined at all. I think it will be important to analyze whether such effect contributes too to the observed phenomenon (similar to the global 3'UTR shortening that occurs during spermatogenesis). Could the authors examine for few relevant genes (preferably from the USPS set), whether the stability of the longer isoform is modulated during the differentiation process (using either cellular model used in the paper for this process)? (relevant to this point, the authors mention that "Notably, genes that showed significant 3'UTR shortening had enriched U-rich motifs in their aUTRs (Fig S8a)" – suggesting involvement of destabilization of the longer isoform during SCT differentiation.

We thank this reviewer for raising this important point. We have carried out further analyses to address this, as listed below:

a) We now show that upregulated genes tend to display greater 3'UTR shortening than no change or downregulated genes (Fig. 5a).

b) As a showcase, we now show isoform stability difference between the two *DNAJC3* isoforms (Fig. 5f). As shown, the short isoform is substantially more stable than the long isoform.

c) We have done more rigorous comparison of sequence motifs in the aUTRs removed by 3'UTR shortening and those related to mRNA stability (Fig. 5g).

d) We now show 3'UTR shortening of USPS genes in multiple models (Fig. S8), which highlights the important consequence of 3'UTR shortening for genes involved in protein secretion.

We believe these new results greatly strengthen the notion that 3'UTR shortening leads to more stable mRNAs, facilitating upregulation of gene expression important for secretory cells.

2. Page 6, 3rd line: Fig. S3b is argued to present the consistency of APA regulation between the hESC and BeWo models. However, the effect shown in this Fig is weak. Could the authors also present a scatter plot with RED values measured in these two models. This will provide a better way to assess the overall consistency.

Thanks for this excellent suggestion. We have added a scatter plot (Fig. 2g) to show consistency of APA between the two models.

3. (a) For all Figs showing RED scores (main and supplementary), indicate in the Y-axis label (or in the legend) what was the reference level (for example, this is clearly indicated in Fig 2f but not 2d). (b) For all

plots (main and supplementary) showing CDF (e.g., Fig3c), please add an indication for the number of genes in each gene set.

Thanks for these excellent suggestions. We have modified our figures accordingly. We have either added gene numbers directly to figures or mention them in figure legends.

4. Fig 4e-f show calculation of r based on only 4 data points. This measure is very sensitive for such a small n (adding one additional point is likely to drastically change the value of r), and therefore the indicated value is not really meaningful. I think that simply showing the plot for the observed tendency without specifying r is better. Same for Fig S12 that is based on only 3 points. Again, r in such case is not really meaningful.

Thanks for this excellent thought. We understand that the meaning of r in this context is not so important. However, as a convention for scatter plots and for completeness, we would like to keep r values in the plots. We do not, however, emphasize their meanings in the text. Hope this is OK.

5. Could the authors color the USPS genes in a different color in Fig 5?

This figure has been removed.

6. p11 – “we found an enrichment of genes whose expression levels were negatively correlated with 3’UTR size”. “enrichment” by which statistical test? Pval?

Sorry for the confusion. The “enrichment” referred to the bottom 5% of all genes based on correlation coefficients (correlation between expression level and 3’UTR RED). We have now clarified this in the figure legend.

7. Fig S13. Why these specific genes are shown? Please explain in a legend.

In Fig. S13a our intention was to show example genes of different types, i.e., positively correlated, negatively correlated, or not correlated between expression level and 3’UTR RED. This is for illustration purpose only. We have now made this point more clear in the figure legend (now Fig. S11a).

REVIEWERS' COMMENTS:

Reviewer #1 (Remarks to the Author):

The manuscript is now easier to read, but I still find the story being told unclear. The take home message, as I see it, is essentially that in two secretory systems (syncytiotrophoblast and plasma cells) protein secretion is made more robust by extending the half life of transcripts encoding the proteins involved in the machinery of secretion. If I'm correct, something like that might be incorporated into the summary.

Also, going back to the choice of USPS genes (based on their statistical significance) what do PLEKHA6 and TIMP2 have to do with protein secretion?

Reviewer #2 (Remarks to the Author):

I am satisfied with the authors responses. At this point, I am supportive of the manuscript getting accepted.

Reviewer #3 (Remarks to the Author):

The authors have addressed well the points I had raised. I have no further concerns.

I spotted two typos:

Page 6: Both 3'READS an RNA-seq data for an example gene DNAJC3. Should be "and RNA-seq"

Page 6: and IPA activation (a 20.9-fold bias in gene number, Fig. 4c). Should be "Fig S4c".

Point-by-point response to reviewers' comments

Reviewer #1 (Remarks to the Author):

The manuscript is now easier to read, but I still find the story being told unclear. The take home message, as I see it, is essentially that in two secretory systems (syncytiotrophoblast and plasma cells) protein secretion is made more robust by extending the half life of transcripts encoding the proteins involved in the machinery of secretion. If I'm correct, something like that might be incorporated into the summary.

We have further clarified this point in our abstract.

Also, going back to the choice of USPS genes (based on their statistical significance) what do PLEKHA6 and TIMP2 have to do with protein secretion?

These two genes were included to serve as technical validation of our sequencing result. We did not intend to choose USPS genes only. Please note many genes other than those in secretion pathways also displayed 3'UTR shortening (indicated in the GO result in Table 1).

Reviewer #3 (Remarks to the Author):

The authors have addressed well the points I had raised. I have no further concerns.

I spotted two typos:

Page 6: Both 3'READS an RNA-seq data for an example gene DNAJC3. Should be "and RNA-seq"

Fixed.

Page 6: and IPA activation (a 20.9-fold bias in gene number, Fig. 4c). Should be "Fig S4c".

Fixed.